# Pervasive misannotation of microexons that are evolutionarily conserved and crucial for gene function in plants

Huihui Yu [1], Mu Li[1], Jaspreet Sandhu[2], Guangchao Sun[2], James C. Schnable [2,3], Harkamal Walia[2,3], Weibo Xie [4✉], Bin Yu [1,3✉], Jeffrey P. Mower [2,3✉] & Chi Zhang [1,3✉]

It is challenging to identify the smallest microexons (≤15-nt) due to their small size. Consequently, these microexons are often misannotated or missed entirely during genome annotation. Here, we develop a pipeline to accurately identify 2,398 small microexons in 10 diverse plant species using 990 RNA-seq datasets, and most of them have not been annotated in the reference genomes. Analysis reveals that microexons tend to have increased detained flanking introns that require post-transcriptional splicing after polyadenylation. Examination of 45 conserved microexon clusters demonstrates that microexons and associated gene structures can be traced back to the origin of land plants. Based on these clusters, we develop an algorithm to genome-wide model coding microexons in 132 plants and find that microexons provide a strong phylogenetic signal for plant organismal relationships. Microexon modeling reveals diverse evolutionary trajectories, involving microexon gain and loss and alternative splicing. Our work provides a comprehensive view of microexons in plants.

[1] School of Biological Sciences, University of Nebraska, Lincoln, NE 68588, USA. [2] Department of Agronomy and Horticulture, University of Nebraska, Lincoln, NE 68583, USA. [3] Center for Plant Science Innovation, University of Nebraska, Lincoln, NE 68588, USA. [4] National Key Laboratory of Crop Genetic Improvement, Huazhong Agricultural University, 430070 Wuhan, Hubei, China. ✉email: weibo.xie@mail.hzau.edu.cn; byu3@unl.edu; jpmower@unl.edu; czhang5@unl.edu

In most eukaryotes, mRNA splicing is an essential step in gene regulation for biological processes. The presence of introns can affect gene expression[1], while splicing defects can promote disease[2]. Alternative splicing is pervasive in animals and plants, and generates protein diversity[3]. Many cases reported suggest it is important for stress response in plants[4–7]. The discovery of short exons, ≤51 nucleotides (nt), known as microexons[8,9], has revealed additional regulatory roles for introns in many eukaryotes, including vertebrates[9–12], insects[13,14], and plants[15–17]. Alternative splicing of microexons in animals revealed the importance of microexon inclusion or exclusion. For instance, a neural-regulated 6-nt microexon in the nuclear adapter Abpp1 enhanced its interaction with Kat5, and misregulated microexons in brain tissues are associated with autism spectrum disorder[9,10]. In insects, alternative splicing of microexons creates multiple forms of the cell adhesion molecule fasciclin I that amino acids inserted by alternative microexon splicing may alter the binding specificity of fasciclin I[13].

Compared with animals, microexons in plants are less studied. Only a few cases of microexons have been described, such as a 9-nt microexon in the Apetala 2 (AP2) domain in *Arabidopsis*[18], a 9-nt microexon in invertase mRNAs in potato[19], a 1-nt microexon in *APC11* (anaphase-promoting complex subunit 11) gene in *Arabidopsis*[15], and a 9-nt microexon in the sucrose fructosyltransferase gene in wheat[20]. Among plant genomes, only a few genomes have been systematically screened for microexons, such as rice[16] and cotton[17], in which ~8000 internal microexons were identified from RNA-seq data and some of them exhibit alternative splicing. Especially, RNA-seq analysis integrating multiple sequencing technologies identified a tissue-specific alternative splicing of a 45-nt microexon located within the AP2 domain of RAP2-7 protein, which fine-tunes DNA binding activity in cotton[17]. However, these works defined microexons by using 51-nt as the minimum length cutoff and did not pay special attention to the smallest microexons (1–15 nt), which were usually missed in genome annotations and transcriptome studies, leading to challenges to correctly predicting the function of corresponding proteins. Thus, in plants, the abundance and locations of small microexons, their effect on protein sequences, and their shared or unique evolution in plants are poorly understood.

The splicing of microexons was studied in animals, especially in neurogenesis[10,21]. The inclusion of microexon depends on the level of cell specificity and development and is mediated by *cis*-regulatory elements, such as exonic splicing enhancers (ESEs) and intronic splicing enhancers (ISEs)[22]. Compared to regularly sized exons, microexons have less efficiency in splicing because the splicing machinery cannot simultaneously assemble at both the 5′ and 3′ splice sites[23]. However, compensatory mechanisms do exist to favor microexon inclusion[8,23,24]. In the human brain, constitutively spliced (CS) microexons were shown to have strong genomic signatures predicted to facilitate splicing, such as stronger splice-site motifs and a higher density of ISEs compared to longer exons, while alternatively spliced microexons have enriched uridine and cytidine elements located 10–20 nt upstream of the 3′ splice site[9]. In human and mouse neurons, nSR100/SRRM4 was found to regulate the splicing of microexons in neurogenesis and the regulation of microexon inclusion or exclusion plays an important role in neuronal development[10]. Plants may have a different set of genes and mechanisms to regulate the splicing of microexons for at least two reasons. First, plants have fundamentally different tissue types compared with animals and there are unlikely to be homologs to the animal neurogenesis factor such as nSR100. Second, the most prevalent form of alternative splicing in plants is intron retention, whereas in animals it is exon skipping. Recently, widespread intron retention in plants was discovered

to be tightly related to post-transcriptional splicing (PTS) of nascent mRNA[25], contrasting with the typical process of co-transcriptional splicing (CTS) of most plant introns[25–27]. It has been shown that many of the full-length polyadenylated chromatin-bound mRNA molecules still contain a fraction of unspliced introns in the nucleus that eventually get spliced out based on their absence from cytoplasmic transcripts[25]. These transcripts with delayed splicing carry so-called detained introns[28] that require PTS[25]. It is not known whether microexons are associated with detained introns in plants.

Although microexons exist in many species, their small size poses difficulties for identification using standard RNA sequence mapping approaches. During genome annotations, standard annotators adopt a statistical model, such as Hidden Markov Model (HMM), to predict genes and exons, which require a training data set and a minimum sequence, such as three codons, for prediction[29,30]. These requirements do not favor microexon identification. Thus, specialized software tools were developed to identify microexons by correcting EST/cDNA misalignment (Volfovsky's method[22]; GMAP[31]), reducing the size of alignment seeds (OLego[32], ATMap[9]), using OLego to map reads unmapped by STAR (FINDER[33]), annotation-guided microexon discovery (MicroExonator[12]), or comparing to a species-specific alternative splicing database (VAST-TOOLS[34]). VAST-TOOLS[34] is a widely used tool for microexon quantification in animals, but in plants, only *Arabidopsis* has this type of alternative splicing database and the microexons were not specified[35].

Misannotated microexons lead to incorrect gene and protein annotations, which can be an obstacle for biological studies. Thus, more advanced algorithms are needed to better detect microexons in plant genomes. Here, we develop computational pipelines to discover and predict microexons in diverse plants and experimentally validate a subset of microexons via RT-PCR followed by sequencing. Importantly, the inclusion of these discovered microexons improves gene annotations in diverse plant genomes, paving the way for improved implementations of gene annotations and functional genomic studies.

## Results

**Discovery of microexons in plant genomes.** Microexons have been variously defined as ≤15-nt[10], ≤30-nt[8], and ≤51-nt[9] in length. In this study, we considered the smallest microexons (1–15 nt) that are most likely to be missed in genome annotations and transcriptome studies. As in previous studies, we also focused on internal microexons (Supplementary Fig. 1a, b). To develop a microexon discovery pipeline for plant genomes, we collected $1 \times 50$ bp and $2 \times 100$ bp Illumina sequencing data sets from *Arabidopsis* and rice (*O. sativa*) for testing, requiring RNA-seq reads to map fully across small microexons (1–15 nt) and extend across splice junctions by at least 6-nt into each flanking exon (Fig. 1a). We initially evaluated these data sets with two commonly used read-mapping programs, HISAT2[36] and STAR[37], and one *de novo* spliced mapping tool, OLego[32]. The comparison demonstrated that OLego was substantially better than HISAT2 and STAR at the condition of no annotation (ab initio), whereas STAR was better than HISAT2 and OLego at annotation-guided microexon discovery (Fig. 1b). From these initial results, we developed a pipeline that combined additional splicing junctions identified by OLego with existing genome-annotated junctions, which was used to guide STAR or HISAT2 mapping. This combined approach increased microexon identification for STAR and HISAT2, and the combination of STAR and OLego had the best performance of all approaches (Fig. 1b).

Based on these preliminary results, we developed a pipeline to identify short (1–15 nt) internal microexons from plant RNA-seq

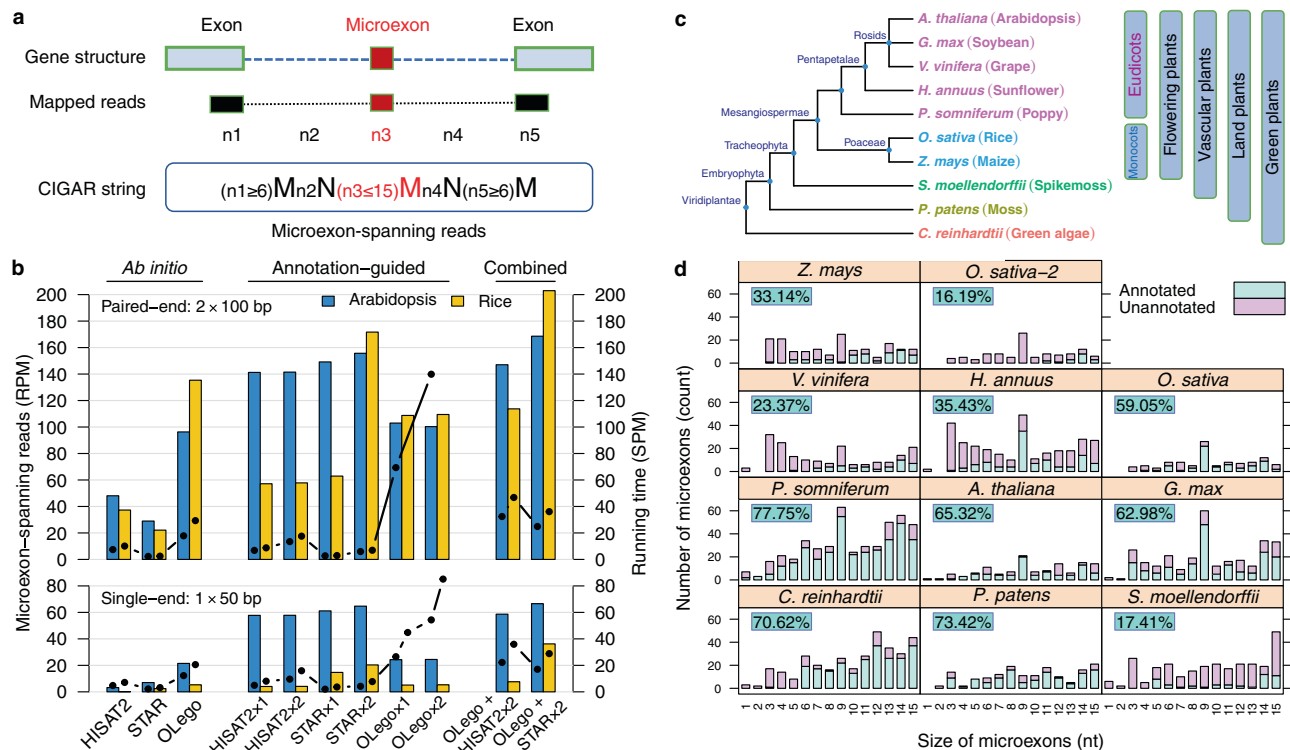

**Fig. 1 Comparison of different methods for microexon detection and summary of microexons identified from RNA-seq in 10 representative plant species. a** Scheme of defining microexon-spanning reads in method comparison. Microexon-spanning reads must be uniquely mapped gap reads with at least five mapped parts (n1–n5: n1 is the part of read aligned to the 5′ flanking exon, n2 and n4 are two gaps, n3 is the part of read mapped to the internal microexons, and n5 is the part of read aligned to the 3′ flanking exon) and each part has a range of nucleotides in CIGAR string of the mapped BAM file (M, an alignment match to the reference; N, a skipped region (e.g., intron) from the reference). **b** Result of microexon detection with different methods on RNA-seq datasets from *Arabidopsis* and rice (*O. sativa*). Bars indicate the number of microexon-spanning reads (left y-axis) and the points in lines indicate the running time (right y-axis). Two samples of 2 × 100 bp parried-end (top) and two 1 × 50 bp single end (bottom) were averaged for plotting, respectively. RPM, reads per million total reads. SPM, seconds per million reads. Method ending with "×1" indicates one-pass, using one round of mapping and "×2" indicates two-pass, using two rounds of mapping. The running time for "OLego×1" (SPM = 332.56) and "OLego×2" (SPM = 666.57) in rice was not shown due to the values >200. **c** Phylogenic tree of the 10 plant species and their groups. **d** The size distribution of microexons identified from RNA-seq data. The percentages indicate the annotation rates. Source data are provided as a Source Data file.

data. We interweaved OLego and STAR to utilize each of their advantages and applied selection criteria that allow junction reads to span one or both splice junctions but require at least 5 reads spanning each junction (Supplementary Fig. 1). In the case of alternative splicing, we chose the intron with the highest average junction-reads in all RNA-seq data. Using simulations, we compared the true positive rate and false discovery rate of our method with MicroExonator and VAST-TOOLS, which showed that our method has a low false discovery rate (1–2%) and the highest true positive rate (90–97%) (Supplementary Fig. 2a, b). Additionally, a performance assessment of these tools on real RNA-seq data (138 samples)[38] showed that all methods discovered many common microexons, and our method detected more microexon candidates and have the largest number of common microexons with the annotation (Supplementary Fig. 2c). To estimate microexon inclusion, we calculated the percent spliced-in (PSI, the percentage of transcript isoforms that contain the microexon) of each microexon discovered by our method and compared with those calculated by MicroExonator and VAST-TOOLS. Common microexons for all methods have a high microexon-inclusion rate (the median PSI values in the population close to 1) and the unique microexons identified only by a single tool have low splicing levels (Supplementary Fig. 2d).

We applied our method to identify short internal microexons in 10 representative plant species (Fig. 1c) for which a collective

990 RNA-seq datasets are publicly available, varying from 42 to 284 for each species and representing multiple tissues and environmental conditions (Table 1 and Supplementary Data 1). A total of 2398 small internal microexons were identified (105–454 per species) that reside in 0.6–2.6% of all expressed genes, most of which (57–90%) are in coding regions (Table 1). Notably, many of these microexons were poorly annotated, ranging from 17.41% in the lycophyte *S. moellendorffii* to 77.75% in poppy (Fig. 1d and Table 1). As a comparison, the proportion of all annotated internal exons (including regular exons and microexons) identified by our pipeline with the same mapping criteria is much higher, from 51.55% in *S. moellendorffii* to 93.08% in *Arabidopsis thaliana* (Table 1). Thus, microexons are substantially more difficult to annotate than regular exons due to their small size. Moreover, different annotations from a single species may have vastly different numbers of annotated microexons. Among two annotations for *Oryza sativa*, MSU7 has a much lower portion of annotated microexons (16.19%) compared with IRGSP1.0 (59.05%), but a higher portion of annotated regular exons (85.73%) for MSU7 than IRGSP1.0 (79.66%).

Among all small microexons (1–15 nt) that we identified in plants, about 50% have sizes in multiples of 3-nt (Supplementary Table 1 and Fig. 1d). This proportion is much lower than that in animals; for example, ~80% of microexons in humans are multiples of 3-nt[9,10]. In general, among flowering plants, the most

**Table 1 Statistics of microexons in 10 plant species identified from RNA-seq data.**

| Species | Number of RNA-seq samples | Number of discovered microexons (1–15 nt) | Microexons being annotated (%)[a] | Total annotated exons (%)[b] | Microexon-containing genes (%)[c] | Coding microexons (%)[d] | Genome annotation version |
|---|---|---|---|---|---|---|---|
| *C. reinhardtii* | 52 | 337 | 70.62 | 89.32 | 2.59 | 89.61 | V5.5 |
| *P. patens* | 99 | 158 | 73.42 | 85.71 | 0.92 | 72.78 | V3 |
| *S. moellendorffii* | 104 | 270 | 17.41 | 51.55 | 1.27 | 51.48 | V1.0 |
| *P. somniferum* | 42 | 454 | 77.75 | 84.23 | 1.00 | 57.93 | Release 100 |
| *A. thaliana* | 138 | 124 | 65.32 | 93.08 | 0.73 | 85.48 | Araport11 |
| *G. max* | 72 | 289 | 62.98 | 81.05 | 0.90 | 81.31 | V2.1 |
| *V. vinifera* | 70 | 184 | 23.37 | 75.06 | 1.07 | 72.28 | V1 |
| *H. annuus* | 61 | 302 | 35.43 | 76.89 | 0.93 | 56.62 | V1.0 |
| *O. sativa* | 284 | 105 | 59.05 (16.19)[e] | 79.66 (85.73)[e] | 0.61 | 80.00 | IRGSP1.0 (MSU7)[e] |
| *Z. mays* | 68 | 175 | 33.14 | 88.68 | 0.80 | 72.00 | AGPv4 |

[a] The percentage of microexons discovered by our pipeline that were annotated in the reference genome.
[b] The percentage of discovered exons that were annotated.
[c] The percentage of genes containing at least one discovered microexon.
[d] The percentage of discovered microexons that are parts of coding regions in transcripts.
[e] Numbers in brackets are from MSU7 annotation and the outside from IRGSP1.0 for *O. sativa*.

abundant length of microexons observed is 9-nt (Fig. 1d). PSI values (the percentage of transcript isoforms that contain the microexon) of most coding microexons were >0.5, indicating that coding microexons had alternative splicing but were usually included in the major transcript isoforms of parent genes (Supplementary Fig. 3). Therefore, coding microexons may have important functions for gene annotation, but it is as yet unclear whether the minor alternative splicing variants are functionally important or transcriptional splicing noise.

**Microexons have increased detained introns requiring PTS.** The genome-wide discovery of microexons in multiple plant species provides an opportunity to study the mechanisms underlying microexon splicing. Because microexons are too small to accommodate ESEs, their ISEs may play a larger role in splicing regulation. The software tool MEME[39] found that T-rich (U-rich in RNA) and G-rich motifs are enriched in introns surrounding microexons (Fig. 2a). In terms of frequency, 14.1% of introns surrounding microexons have T-rich motifs, while only 9.3% of introns surrounding regular exons have this type of motif (Fig. 2b). G-rich motifs are also enriched being present in 12.1% of introns surrounding microexons compared with only 4.8% for regular exons (Fig. 2b). Wang et al[40]. classified ISE (ISE-hex3) into six groups (A-F), and the group D motif contains the highest frequency of G. In all introns flanking microexons, 44.2% have the group D motif (ISE-hex3-D), while only 35.9% for regular exons (Fig. 2b). The difference is statistically significant. Therefore, the introns flanking microexons tend to have strong intronic splicing signals.

For all discovered microexons in 10 plant species, we found that the proportion of intron retention (i.e., an unspliced intron located within a longer exon of another transcript) is significantly higher around microexons than for regular exons based on RNA-seq data from total mRNA (Fig. 2c), indicating an increased number of detained introns that require post-transcriptional splicing (PTS)[25,28]. Although long-read sequence data are available for studying splicing of nascent RNA[25], long reads could not be used to identify microexons due to their high error rate. Therefore, we applied our microexon-discovery pipeline to a short-read RNA-seq dataset of polyadenylated RNAs bound on the chromatin (CB), in the nucleoplasm (NP), and in the cytoplasm (Cyto), respectively[25]. We found that, among 69

constitutively spliced microexons, >72% (50) have at least one flanking intron that has an increased tendency for delayed splicing; that is, these detained introns exhibit read unspliced ratio >0.1 in CB but close to zero in NP and Cyto. While in the whole genome, only about 30% of introns are detained introns that require PTS[25]. Furthermore, upstream introns (5′ intron) of microexons are more likely to be detained than downstream flanking introns (3′ introns) (Fig. 2d), whereas for ~70% of regular exons, the upstream intron was spliced first during transcription[25]. This finding is consistent with a previous study for a 9-nt microexon in potato invertase[19]. After the further analysis of gapped reads (connecting exon-exon reads) around microexons, we found 16 out of 69 microexons had reads supporting the case of sole-downstream-intron splicing and the case of both-flanking-introns splicing but no reads supporting the case of sole-upstream-intron splicing (see Fig. 2e for an example of this case), another four microexons had reads supporting sole-upstream-intron splicing and both-flanking-intron splicing but no reads supporting sole-downstream-intron splicing, and the remaining 49 microexons had all three types of gapped reads supporting sole-upstream-intron splicing, sole-downstream-intron splicing, and both-flanking-introns splicing, respectively, but most of them (32 microexons) had more reads supporting downstream-intron splicing than upstream-intron splicing. These data indicate that the two flanking introns around microexons are likely removed consecutively rather than simultaneously with transcription, and the upstream introns tend to be detained and have an increased reliance on PTS. Based on this analysis, we proposed a model for microexon splicing in plants (Fig. 2f). During transcription, flanking introns surrounding the microexon in the majority of transcripts are spliced out, but there are still a fraction of transcripts containing unspliced introns, especially for the upstream introns. Subsequently, transcripts undergo post-transcriptional modifications (e.g., polyadenylation) and the detained upstream introns are spliced by PTS, and then the mature mRNAs are released into the cytoplasm. This model suggests that the splicing of introns adjacent to microexons needs a longer time than that of regular exons.

**Microexons encode important protein motifs.** In all 10 plant species, the most common protein motifs encoded partly by microexons (see Methods) were AP2 (Apetala 2, PF00847),

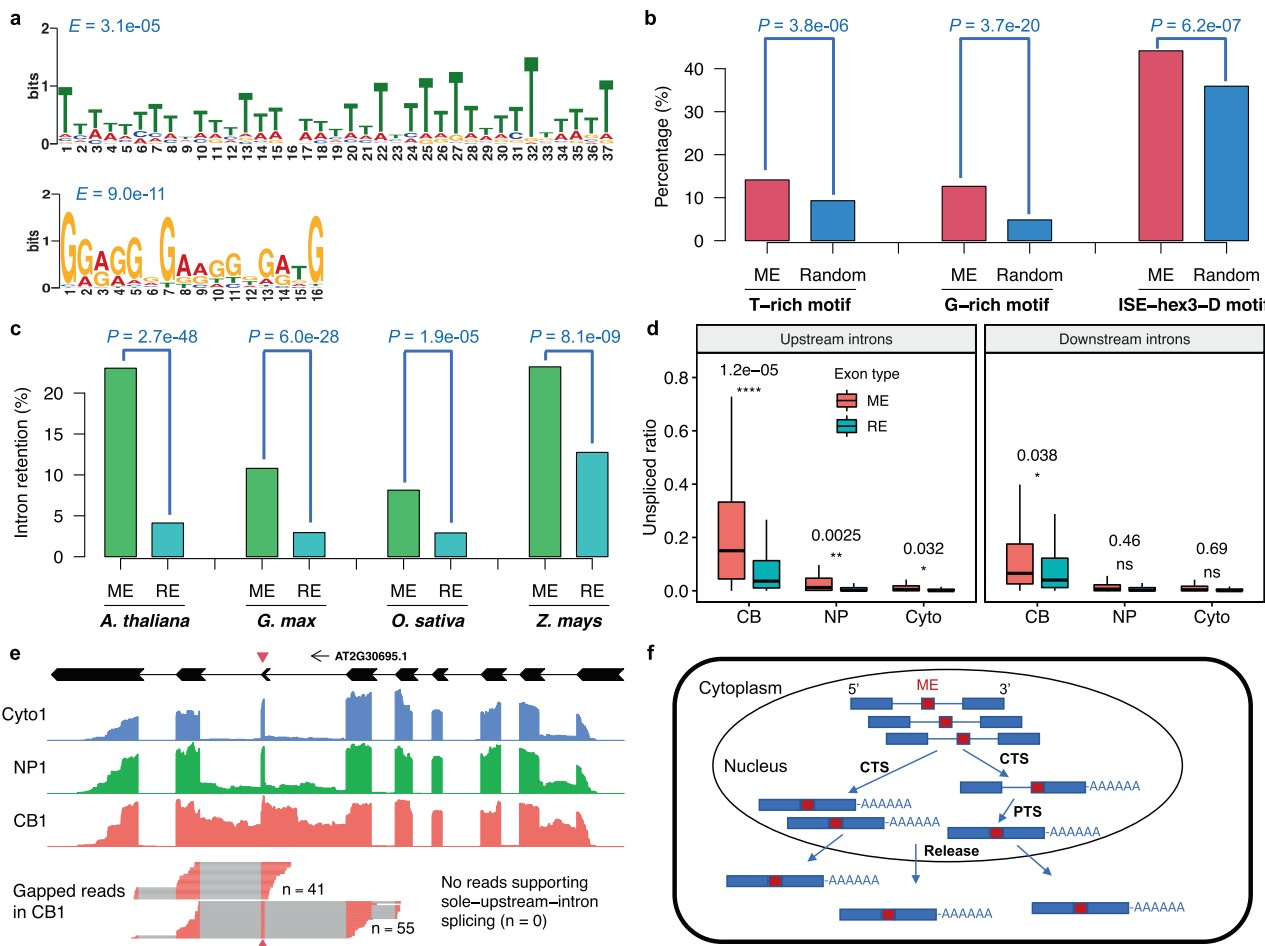

**Fig. 2 Intron splicing motifs and post-transcriptional splicing for microexons. a** Enrichment of T-rich and G-rich motifs in flanking introns of microexons. **b** Comparison of intron splicing motifs in introns surrounding microexons (ME) and randomly selected introns surrounding regular exons (Random). ISE-hex3-D, intronic splicing enhancer hexamer set group D motifs as suggested by Wang et al.[40]. **c** Comparison of intron retention rate in introns surrounding microexons (ME) and introns surrounding regular exons (RE) in four plant species. In **b** and **c**, Pearson's Chi-squared tests were used, and P-values are displayed. **d** Comparison of unspliced ratios for introns surrounding microexons (ME) and regular exons (RE) of polyadenylated transcripts in the chromatin (CB), the nucleoplasm (NP), and the cytoplasm (Cyto) in *Arabidopsis*. The intron unspliced ratio was estimated based on the percentage of intron retention (PIR) value (see Methods section). For each intron, ratios from three biological replicates were averaged. For the box plots, the bounds of a box show the interquartile range (IQR), the center line in the box shows the median, and the whiskers extend to no further than $1.5 \times$ IQR from the box bounds. The significances were calculated by two-sided $t$-tests on the difference between ME and RE flanking introns: ns, $P > 0.05$; *, $P \le 0.05$; **, $P \le 0.01$; ***, $P \le 0.001$; ****, $P \le 0.0001$. **e** An example of the post-transcriptional splicing (PTS) introns surrounding a 14-nt microexon of the *Arabidopsis* gene *AT2G30695*. Beneath the gene structure diagram, RNA-seq read depths are shown in the first replicate of cytoplasm (Cyto1), nucleoplasm (NP1), and chromatin (CB1) samples, respectively. There are reads supporting both-flanking-intron splicing ($n = 55$) and sole-downstream-intron splicing ($n = 41$) whereas no reads support the transcript that has the sole-upstream-intron splicing ($n = 0$) in CB1 sample (SRA: SRR10538411). The red triangles point to the microexons. In **d** and **e**, Illumina RNA-seq data were obtained from Jia et al.[25]. (SRA: PRJNA591665). **f** A model for microexon splicing. Flanking introns of the microexon in the majority of transcripts are spliced out by co-transcriptional splicing (CTS), but there are a fraction of transcripts containing detained introns, especially upstream introns, in the nucleus that require post-transcriptional splicing (PTS) before the mature mRNAs are released into the cytoplasm. Source data are provided as a Source Data file.

Glyco_hydro_32N (Glycosyl hydrolases family 32 N-terminal, PF00251), Myosin_head (Myosin head motor domain, PF00063), bHLH-MYC_N (bHLH-MYC and R2R3-MYB transcription factors N-terminal, PF14215), Peptidase_M1 (Peptidase family M1 domain, PF01433), and Gelsolin (Gelsolin repeat, PF00626), which represent 7.6%, 5.5%, 2.4%, 1.7%, 1.4%, and 1.3% of all coding microexons, respectively (Supplementary Table 2). The AP2 and Glyco_hydro_32N motifs are encoded by 9-nt microexons, which causes the spike in the abundance of 9-nt microexons in flowering plants (Fig. 1d). This distribution of motifs encoded by microexons is primarily representative of flowering plants, with slight variation among more distantly related species.

For example, the top three motifs in the unicellular green alga (*C. reinhardtii*) include Pkinase (Protein kinase, PF00069), PK_Tyr_Ser-Thr (Protein tyrosine and serine/threonine kinase, PF07714), and Myb_DNA-binding (Myb-like DNA-binding, PF00249), while in moss (*P. patens*) and spikemoss (*S. moellendorffii*) the top motifs are bHLH-MYC_N, Myosin_head, and Peptidase_M1.

Those microexon-coding proteins have important functions in plants. For example, in flowering plants, duplication of AP2 and invertase genes containing 9-nt microexons enables broad environmental adaptation[41,42]. AP2 domain-containing proteins belong to the AP2/EREBP superfamily, which is a large

superfamily of transcription factors with four major families: AP2, RAV, DREBs, and ERFs, and AP2 family proteins contain one or two AP2 domains[43,44]. It has been shown that genes in the AP2 family are involved in the regulation of developmental processes, such as organ number and size control, shoot and root meristem maintenance, flower initiation, and growth[45]. In AP2 family genes, the 9-nt microexons encodes amino acid sequence VYLG (phase 1), which is located within the core of the first AP2 DNA-binding domain[46]. Nearly all the 9-nt microexons in AP2 genes were not annotated in rice by MSU7 (none annotated) and maize by AGPv4 (only one annotated), resulting in the incorrect functional annotation of more than 15 proteins in each species. Glyco_hydro_32N is the N-terminal domain of glycoside hydrolase family 32, which is a family of glycoside hydrolases. Glycoside hydrolase family 32 contains invertase (β-fructofuranosidase, EC 3.2.1.26) and other fructofuranosidases[47]. Invertases irreversibly hydrolyze sucrose into glucose and fructose and are involved in regulating plant carbohydrate partitioning, developmental processes, hormone responses, and biotic and abiotic interactions[48,49]. In invertase genes, the 9-nt microexon encodes D-P-N/D-G/A (phase 1), a part of a beta-fructosidase motif[19,20]. None of these 9-nt microexons in invertase genes were annotated in rice and maize, causing the functional annotation of 7 proteins to be incorrect in each species. Therefore, protein models for important regulatory and developmental factors were substantially improved because of the discovery of microexons.

**Microexons are crucial for plant genome annotation**. Because the pervasive misannotation of microexons in plant genomes often results in incorrect gene models and protein sequence inferences, we next sought to examine the effect of these misannotated microexons on the plant genome annotation. We compared gene models and inferred protein sequences that included microexons discovered by our pipeline with those from the reference genome annotations for *Arabidopsis*, Soybean, rice (MSU7), and maize. Most coding microexons discovered by our pipeline were not annotated in the reference genomes of rice (81%) and maize (65%), whereas fewer were missed in the *Arabidopsis* (27%) and soybean (23%) genome annotations (Supplementary Table 3). Some of these unannotated microexons overlapped with or were fully located within larger exons annotated in the reference genome annotations, particularly for *Arabidopsis* (explaining 27 of 29 unannotated microexons). These overlapping microexons potentially represent alternative splicing variants or misannotations in the reference genome. Some unannotated coding microexons discovered by our pipeline were located within genes that were either completely unannotated or annotated as non-coding genes in the reference genome annotations (Fig. 3a and Supplementary Table 4). The remaining coding microexons were discovered within introns of existing gene annotations and generally improved predicted protein length (Fig. 3a and Supplementary Table 4), as the lack of inclusion of these microexons resulted in truncated protein sequences and long 5′ or 3′ untranslated regions (UTRs) due to frameshifts in the gene models and the usage of early stop codons (mainly in rice; Supplementary Fig. 4), or deletion of a few amino acids in protein sequences due to microexon-skipping in the gene models (mainly in soybean and maize; Supplementary Fig. 5). A small number of cases had no change in protein length, which was caused by an incorrect extension of an adjacent exon, often accompanying the usage of a non-canonical intron-exon junction (i.e., not GY-AG). For example, AP2 family gene *OsWRI1-1* (*LOC_Os11g03540*) was reported to have an atypical splicing junction[50], but it is actually an incorrect annotation that missed a 9-nt microexon. Importantly, the reference gene models

produced protein sequences with fewer known motifs on average than gene models including microexons (Fig. 3b, c). This was particularly problematic for rice and maize, as some AP2 family proteins could not be annotated due to the missed annotation of 9-nt microexons (Fig. 3d).

In addition to issues arising from missing microexons, a large number of annotated microexons in the four reference genomes were not supported by the many RNA-seq data sets used by our pipeline (Supplementary Table 3). Again, this was most notable for the two monocots, as nearly all microexons annotated in their reference genomes (91.3% for rice and 92.4% for maize) did not have supporting junction reads. The majority of microexons annotated in soybean were also unsupported (60%). Even in the highly curated *Arabidopsis* reference genome, 37% of annotated microexons were unsupported. These unsupported microexons did not have sufficient quality-junction-reads from RNA-seq data, which cannot be attributed to mapping errors because STAR is sensitive to genome annotation and will correctly map reads to microexons if the annotation is correct. These unsupported microexons are also unlikely due to low expression based on the extensive coverage and tissue variability of the RNA-seq data sets used. Instead, many of these unsupported microexons are false positive exons predicted during annotation. For example, the gene *AtRLP1* has a gene model that has three extra exons, one of which is a 13-nt microexon, and these extra exons have no RNA-seq read supporting while the flanking exons have clear RNA-seq read coverage (Fig. 3e).

**Prediction and verification of additional plant microexons**. The RNA-seq pipeline detected many unannotated microexons, but it may miss some microexons because it cannot be used for species that lack sufficient RNA-seq data. To find additional coding microexons in plants that may have been missed by our RNA-seq pipeline, we developed a predictive modeling approach. Due to the small size of microexons, we increased the signal by defining a microexon-tag, which includes the coding microexon and portions of flanking exon sequences (Fig. 4a). Microexon-tags were grouped according to microexon sizes and phases, and microexon-tags in each group were clustered based on pairwise alignment scores of translated amino acid sequences (Fig. 4a). Thus, in each cluster, the microexons have the same size and phase, and the microexon-tags are highly similar in coding and peptide sequences. After testing the effect of microexon-tag size on microexon modeling, we chose a length of 108-nt due to higher accuracy and lower false positives (Supplementary Fig. 6).

Based on 108-nt and the translated 36 amino acid (aa) microexon-tags, a total of 45 clusters of microexon-tags were discovered, with the requirement that each cluster must contain at least three of nine land plant species (*C. reinhardtii* was excluded because it shares few microexons with land plants) (Supplementary Table 5). These 45 microexon clusters cover about 70% of coding microexons in plants (e.g., 70 of 106 microexons in *Arabidopsis*, 147 of 235 in soybean, 66 of 84 in rice, and 81 of 126 in maize). For these 45 microexon-tag clusters, microexons ranged in size (1–15 nt) and phase (a total number of 16, 21, and 8 in phase 0, 1, and 2, respectively), and microexon-tags always spanned at least 3 exons (and occasionally up to 5 exons when the flanking exons were short). Based on the set of 45 clusters of conserved microexon-tags, we developed a microexon modeler in plant genomes (MEPmodeler[51]) to predict microexons for plant species without using RNA-seq data and genome annotations. MEPmodeler[51] scanned genome sequences to identify microexon-tag location exhibiting not only sequence similarity of the coding exons, but also the structure of microexon-tag, including the positions and phases of microexons

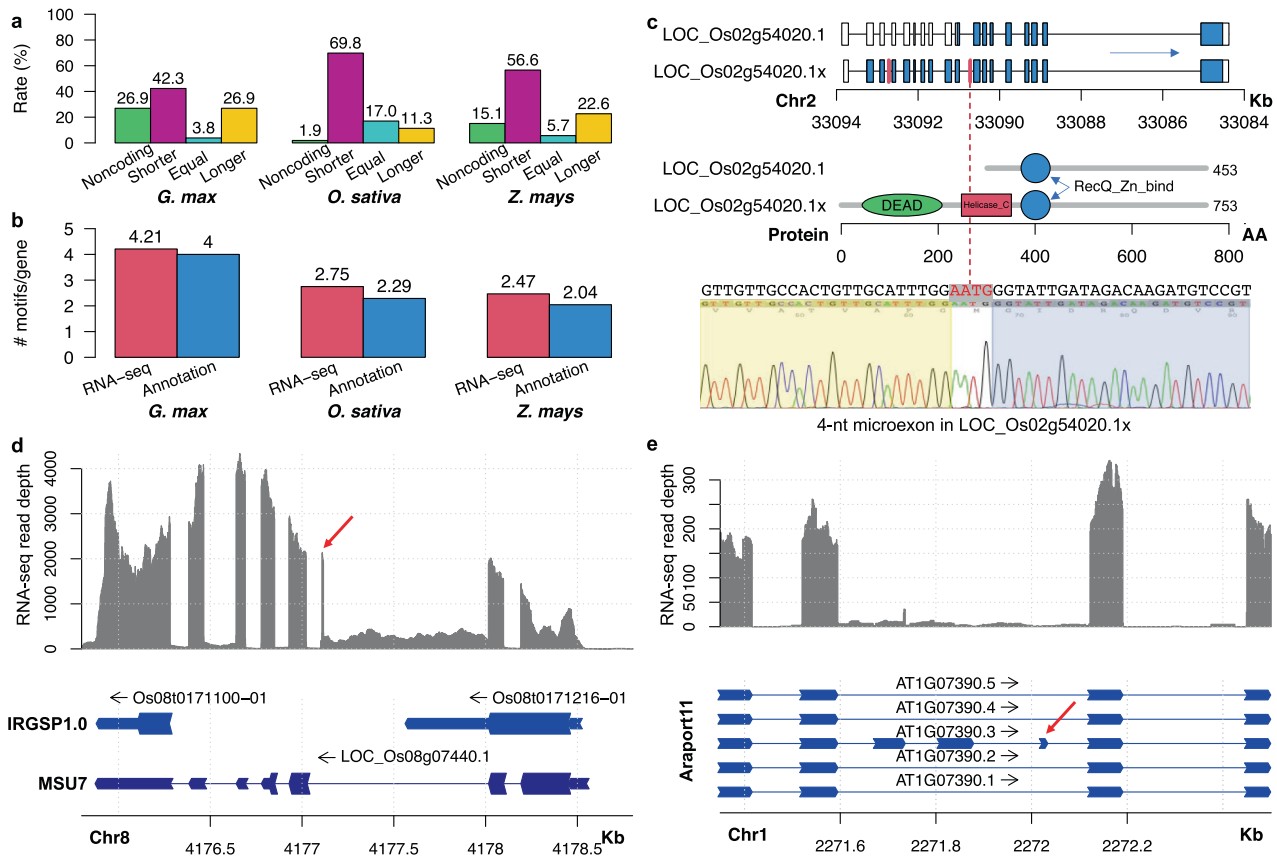

**Fig. 3 Comparison of annotated microexons and discovered microexons from RNA-seq data. a** The comparison of protein lengths from genome annotations with missing microexons to the protein lengths with considering discovered microexons. **b** Comparison of the number of protein domains encoded by transcripts containing microexons and annotated transcripts with missing microexons. **c** An example of a 4-nt microexon not annotated in rice. An annotation lacking this microexon causes a long 5′ untranslated region and a truncated protein with two important protein motifs lost. This microexon was validated by our RT-PCR experiment (see Methods section). LOC_Os02g54020.1 is the annotated transcript and LOC_Os02g54020.1x is the assembled transcript from RNA-seq data. **d** An example of a 9-nt microexon that was not annotated by MSU7 and IRGSP1.0 in rice, resulting in discordant annotations in MSU7 (one gene) and IRGSP1.0 (two genes). **e** An example of an annotated microexon in gene *AtRLP1* (*AT1G07390*) by Araport11 that lacks RNA-seq read support in *Arabidopsis*. Red arrows point to microexons.

and flanking intron sizes. First, the nucleotide sequences of 45 microexon-tag clusters were scanned against the NCBI plant EST (Expression Sequence Tag) database to calculate a Position Weighted Matrix (PWM) of nucleotide sequences in each microexon-tag cluster based on an intronless RNA-derived data set (Fig. 4a). Next, we developed a method to model microexons in genome sequences, by using the PWM scores and also allowing gaps for introns on any side of the microexon but keeping the positions of microexons and phases conserved (see Methods). Multiple sequence alignments revealed strong conservation in microexon position and overall structures of parent genes, especially in flowering plants (Supplementary Figs. 7–51). Some clusters are located in the same family of genes. For example, Cluster 7 (size: 5; phase: 1; Supplementary Fig. 13) and Cluster 28 (size 12; phase 0; Supplementary Fig. 34), are adjacent in Peptidase M1 proteins (e.g., AT1G63770.5), while Cluster 5 (size 4; phase 2; Supplementary Fig. 11) and Cluster 18 (size 8; phase 1; Supplementary Fig. 24) are in distal locations in genes for RECQ3, an ATP-dependent helicase (e.g., AT4G35740.1).

The microexon modeler results were validated in three different ways. First, MEPmodeler[51] was applied to 4 of the 9 land plant species that were previously used for training (*Arabidopsis*, soybean, rice, and maize), and >90% of microexons in these 45 clusters were predicted (Fig. 4b, left). This microexon modeler also discovered microexons that were not identified by

the primary microexon identification pipeline based on RNA-seq data. When the predicted junction information for microexons was included for STAR remapping, about half of these microexons were supported by RNA-seq reads (Fig. 4b, right). For example, the prediction improved the mapping around a 1-nt microexon in soybean (Fig. 4c). Second, we conducted RT-PCR and Sanger sequencing of amplicons covering 10 microexons (size ≤ 10-nt) selected from each of the four species, i.e., *Arabidopsis*, soybean, rice, and maize, and all these microexons were experimentally validated (Fig. 4d and Supplementary Figs. 52–55). Third, MEPmodeler[51] was applied to tomato (*S. lycopersicum*), which was not used for training. As a result, 78 unique microexons were predicted, of which only 38 were included in the genome annotation (version SL3.0) (Fig. 4e). To evaluate the accuracy of these microexon predictions in tomato, we used STAR to map RNA-seq data of three replicates of root tissue to the reference genome and counted microexon-spanning reads (as defined in Fig. 1a). Four different sets of junction information were provided for mapping: no junction (ab initio), annotated junctions (anno), predicted junctions (pred), and both annotated and predicted junctions (anno+pred) (Fig. 4e). The comparison showed that predicted junctions can guide STAR to find more microexon-spanning reads based on reads per million total reads (RPM) than annotated junctions (103.04 RPM v.s. 80.38 RPM), even though the number of predicted microexons

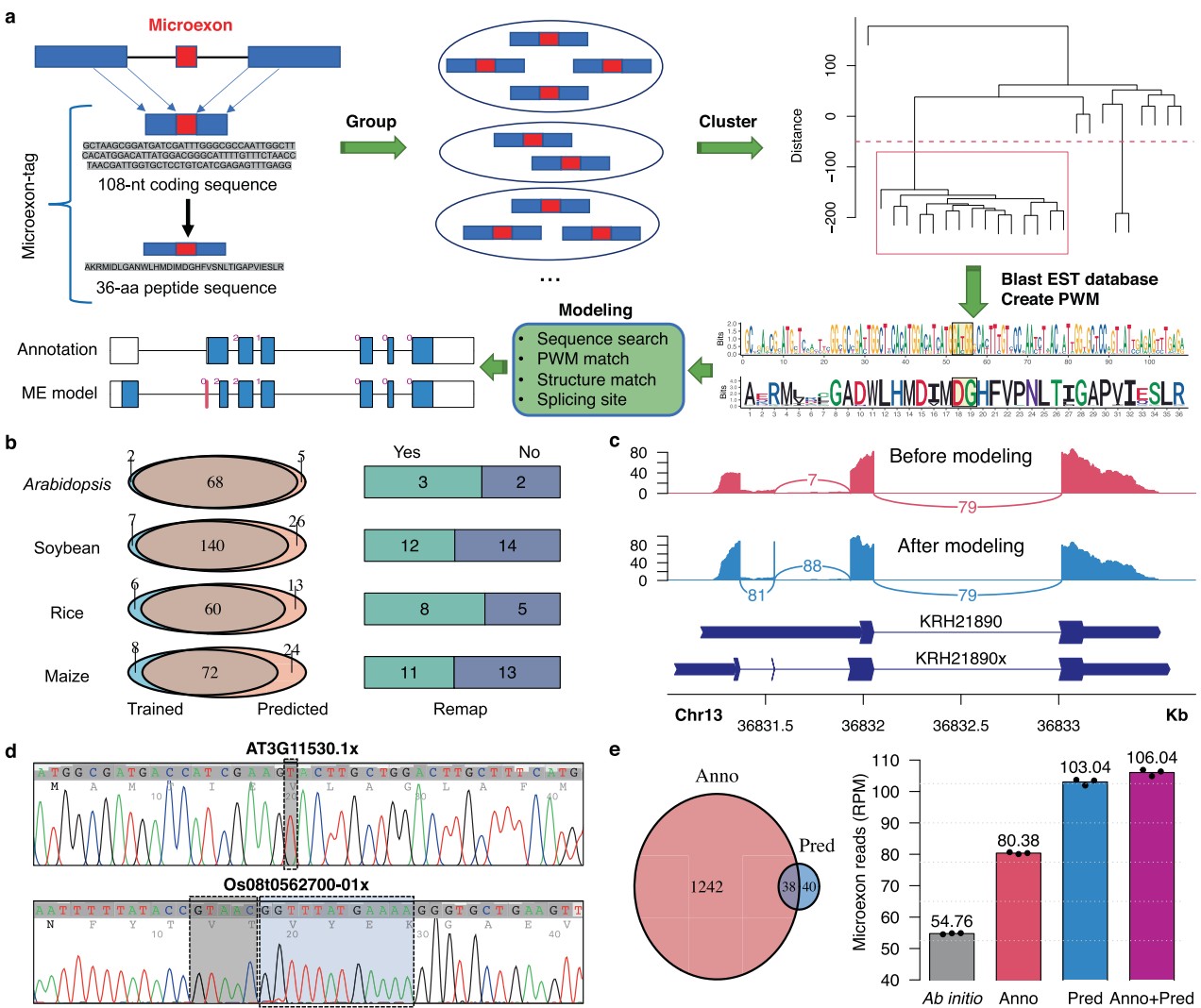

**Fig. 4 Conserved coding microexon modeling and validation. a** Scheme of the pipeline for microexon clustering and prediction. **b** Comparison of predicted and trained microexons. For predicted microexons being not in the training set, some could be identified by remapping all the RNA-seq data with predicted junction information added to the microexon discovery pipeline. **c** An example of improved mapping reads around the 1-nt microexon of *APC11* gene in soybean. Before the 1-nt microexon modeling, only 7 junction reads were mapped spanning the downstream intron of this microexon, whereas 81 and 88 junction reads were mapped spanning the upstream and downstream introns, respectively, when the annotation of this predicted 1-nt microexon was provided. **d** Two examples of validated microexons by RT-PCR sequencing. *Arabidopsis* transcript AT3G11530.1x contains a 1-nt microexon (up) and rice transcript Os08t0562700-01x contains two adjacent microexons (5-nt and 12-nt, bottom). The 1-nt and 5-nt microexons are missing in genome annotations (transcripts end with an 'x' to distinguish from annotations). **e** Comparison of predicted microexons with annotated microexons in tomato (*Solanum lycopersicum*). The left panel shows the numbers of microexons from the genome annotation and the microexon modeling, and the right panel shows the mapping result of STAR on three root RNA-seq datasets with different sources of junctions. Data are presented as bars (mean values) and points (individual values). RPM, reads per million total reads. Ab initio, no junction information provided for mapping; Anno, junction information from the reference genome annotation (SL3.0); Pred, junction information from the result of microexon modeling; Anno+pred, junction information from both the genome annotation and the microexon modeling.

(78) was much smaller than annotated microexons (1,280) (Fig. 4e). When annotated junctions were added to predicated junctions, the number of microexon-spanning reads was not largely increased (103.04 RPM to 106.04 RPM). We can infer that the microexon modeling captured most of the highly expressed microexons, and many of annotated microexons had few RNA-seq reads supporting the annotated junctions. Most microexon-spanning reads belong to microexons modeled by MEPmodeler[51] because modeled microexons had high expression levels in all microexons, which indicates modeled microexons are true positive. These validations indicate that the predicted microexons are reliable and the microexon-modeling algorithm is useful for improving genome annotation.

**Evolution and conservation of plant microexons**. Because microexon-tags contain gene structural information, have fixed lengths of nucleotide and amino acid sequences, and are conserved in land plants, they could serve as potential molecular markers for evolutionary studies. Therefore, we applied the prediction method to 132 land plants and used microexon-tags as a marker to study the dynamics of microexon evolution. We constructed a phylogenetic tree based on consensus string in a cluster (see Discussion and Methods section), which recapitulates land plant relationships including monophyletic groups for eudicots, monocots, magnoliids, asterids and rosids within eudicot, and angiosperms as a whole, which were sister as expected to gymnosperms (Fig. 5). According to the heatmap of predicted

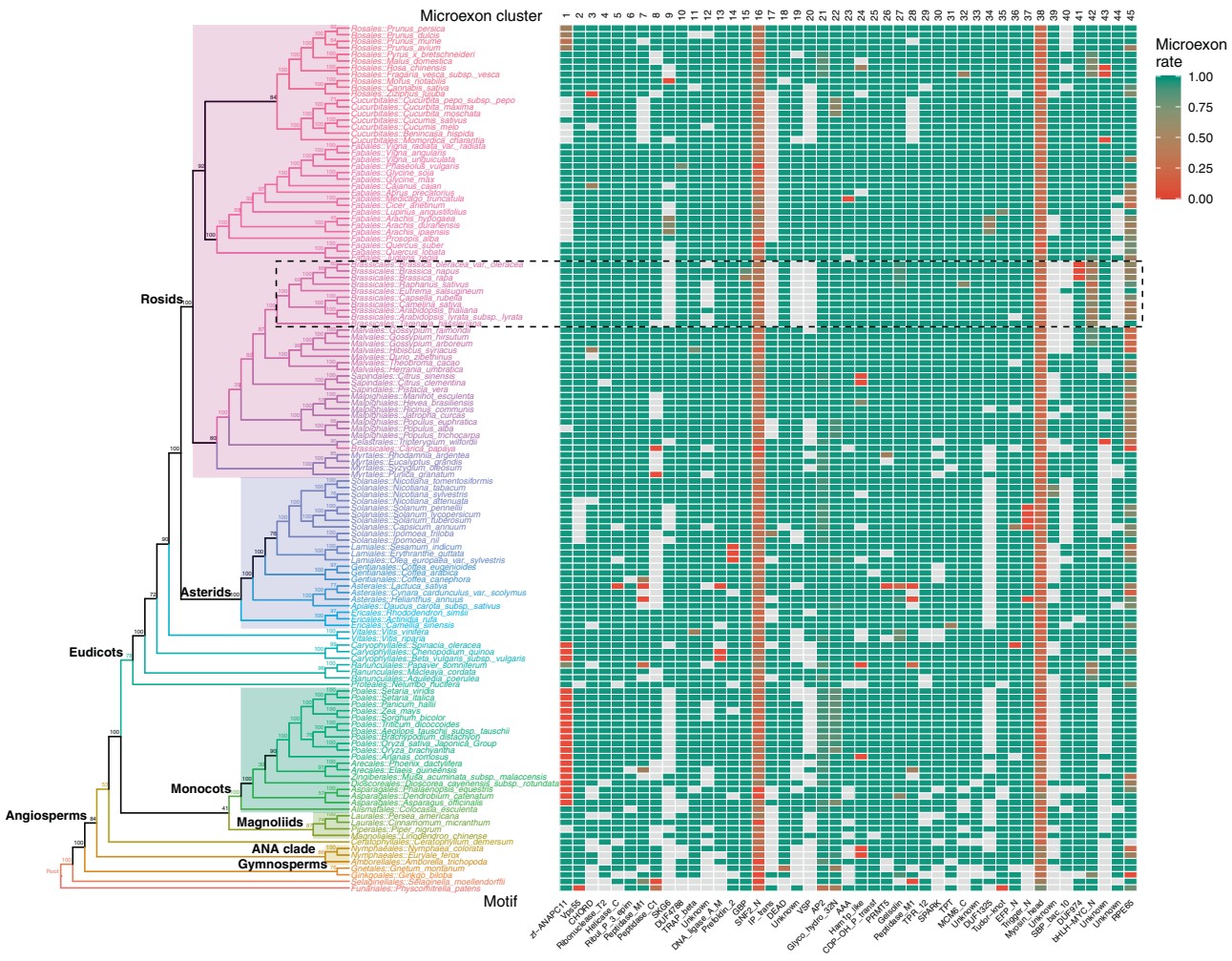

**Fig. 5 Phylogenetic tree of land plant species based on predicted microexon-tags.** Tip labels of the species in the same order have the same color. In the heatmap, microexon rate is the rate of microexons among all predicted results in a given cluster in a given species, e.g., green indicates that 100% microexons are present, red indicates all microexons are missing due to intron loss in the microexon-tags, and the others are between 0 and 1. For example, the first cell is brown for *prunus persica* in Cluster 1, because there are two predicted microexon-tags: one has a microexon with both flanking introns and the other is merged to a long exon (not an actual microexon), and the microexon rate = 0.5. A white cell indicates microexon-tag could not be found (i.e., NA). The dashed rectangle denotes the order with extensive missing predicted microexon clusters. Source data are provided as a Source Data file.

microexons in 132 land plants, some flowering plant groups, such as Brassicales (represented by *A. thaliana*), have experienced extensive sequence diversity and are less predictable (Fig. 5). The heatmap also indicates that microexons in nonflowering plants (moss and spikemoss) are relatively less predictable, which is reasonable because the prediction pipeline was mainly trained on flowering plants for which a lot of EST sequences are available. For Cluster 16 (encoding SNF2 family N-terminal domain, SNF2_N) and Cluster 38 (encoding Myosin head, motor domain), the low frequency of microexons is caused by the presence of two gene-structure variants that have either retained or lost flanking introns in almost every land plant.

Moreover, we found substantial diversity in the timing and rate of retention of microexons during land plant evolution (Fig. 5). Based on multiple sequence alignments of 10 plant species (Supplementary Figs. 7–51), 35 of 45 microexons clusters are shared with flowering plants and moss (but not green algae), suggesting that most (about 80%) extant microexons may date back to the universal ancestor of land plants (Supplementary Table 5). Only four microexon clusters are specific to flowering plants (Cluster 4, 12, 39, and 43), indicating that less than 10% of extant microexons arose relatively recently in a common ancestor

of some or all flowering plants. Another five microexons are shared with flowering plants and *Selaginella* (but not *Physcomitrella*), indicating that ~10% of extant microexons arose early in the evolution of vascular plants. By contrast, only one microexon is shared with *Chlorella* (Cluster 10; Supplementary Fig. 16), suggesting that very few microexons were originated prior to the colonization of land and/or were retained over such a long time. When microexon-tag sequences cannot be recognized as microexons, it is most commonly caused by the loss of flanking introns from the genes. For example, microexons in Cluster 1 were merged into a long exon in most monocots due to the loss of all introns in their genes by reverse transcriptase (RT)–mediated intron loss[52,53] (Fig. 5, and see the following section). Likewise, microexon-tag sequences could not be recognized as microexons due to the loss of flanking introns for *Helianthus* in Cluster 7 (Supplementary Fig. 13), and in many other examples (Fig. 5, red cells). Microexons for *Arabidopsis* in Cluster 40 (Supplementary Fig. 46) and Cluster 44 (Supplementary Fig. 50) represent another pathway for microexon-tag sequences that cannot be recognized as microexons, as microexons were extended to longer exons due to usage of an alternative splice site. Microexons can be lost when the parent gene is completely lost during evolution. For example,

the parent gene in Cluster 17 has been lost from 5 of 10 plant species, including from *Arabidopsis* such that it has no homolog to the gene *Os03t0287100-01*, which encodes a phosphatidylino-sitol transfer protein in rice (Supplementary Fig. 23). In the following section, we will describe a very special case of microexon loss by intron retention in Cluster 2.

**1-nt microexons of *APC11* and *VPS55***. Due to their sizes, <3-nt microexons can be particularly difficult to identify and verify by short read mapping, and they can suffer from high rates of false positive detection. Nevertheless, these microexons exist in plants, such as a 1-nt microexon that was reported and validated for the *APC11* gene in *Arabidopsis*[15]. In our study, two well-conserved clusters of 1-nt microexons were detected by our RNA-seq pipeline and extended prediction approach, including the pre-viously reported microexon in *Arabidopsis APC11* (Supplemen-tary Fig. 7) and a discovered microexon in *VPS55* (Supplementary Fig. 8). In all plant species, these two sets of 1-nt microexons were missing in their reference genome annotations. Annotations lacking these 1-nt microexons predict proteins with shorter or different N-terminal peptides. For example, in flowering plants, the 1-nt microexon is usually not annotated in its host *VPS55* gene, which results in the absence of N-terminal amino acids (13 amino acids in *Arabidopsis*) and the incorrect number of trans-membrane helices and an opposite orientation of the first helix in the gene products – there are four transmembrane helices in the full-length VPS55 protein while only three in annotated protein with the missing microexon (Supplementary Fig. 56). Never-theless, these 1-nt microexons are well conserved in diverse flowering plants and other land plants (*Physcomitrella* and/or *Selaginella*). Furthermore, RNA-seq mapping and RT-PCR sequencing confirmed the 1-nt microexons in *APC11* (Fig. 4c) and *VPS55* (Fig. 4d).

The APC11 protein is a ubiquitin ligase that has essential functions in the cell cycle as a subunit of the anaphase-promoting complex or cyclosome[54]. Plants typically have 1–2 copies of the *APC11* gene, with duplicate copies evolving recently and independently in each species according to the phylogenetic tree (Fig. 6a). RNA-seq data provides strong support for the 1-nt microexon in *APC11*; for example, the remapping result in *G. max* clearly shows the 1-nt microexon while STAR could not map reads correctly without the 1-nt microexon annotation (Fig. 4c). In most land plants, this 1-nt microexon is present and flanked by two introns, whereas in most monocots the gene is intronless and the 1-nt sequence is subsumed in the single large exon (Fig. 6a). In the moss *P. patens*, the *APC11* gene (*Pp3c12_19260*) has at least two alternatively spliced isoforms: one with the microexon and the other with the 1-nt sequence included at the 3′ end of a larger exon. In *P. somniferum* and species in *Prunus* genus, *i.e., P. mume, P. persica, P. avium*, and *P. dulcis*, one copy of the gene contains the microexon and its flanking introns, while other copies are intronless (Cluster 1 in Fig. 5). Several incomplete copies of *APC11* genes also exist in the *P. somniferum* genome, and these incomplete copies do not have this microexon. Because this 1-nt microexon evolved in early plants and is retained in most land plants, we can infer that the intronless genes were a result of RT–mediated intron loss[52,53].

*VPS55* encodes a vacuolar protein sorting-associated protein 55, which is involved in protein transport from endosomes to vacuoles. This protein is generally encoded by at least two gene copies in plants, with duplication events occurring in flowering plants and *Selaginella* (Fig. 6b). In flowering plants, the gene structures are well conserved (Fig. 6b), and one clade of homologs (*VPS55-1*) has a 1-nt microexon while the other clade of homologs (*VPS55-2*) does not have the microexon (Fig. 6b).

*Selaginella* also has two copies of *VPS55* genes that are almost identical, and each copy has two alternative transcripts: one has the 1-nt microexon and the other loses it because of intron retention (Fig. 6c). Thus, both flowering plants and *Selaginella* create two distinct protein sequences due to the presence or absence of the microexon, but they do so in different ways (Fig. 6c). Nonvascular plants, such as the hornwort *Anthoceros angustus*[55] and the liverwort *Marchantia polymorpha*[56], have a single copy of *VPS55* gene, but like *Selaginella*, there are two different transcripts: with or without 1-nt microexon (Fig. 6d and Supplementary Fig. 57), indicating that this scenario was ancestral. From ferns[57], the *VPS55* gene was duplicated to two copies, one with the 1-nt microexon and the other without the microexon, similar to most flowering plants (Fig. 6e and Supplementary Fig. 57). Interestingly, the splice sites of the 1-nt microexons in *VPS55* are well conserved, especially the 6-nt sequences at the splice donor sites (GTRAGT) (Fig. 6d). During evolution, these two genes with the 1-nt microexon gained a stronger splicing site, i.e., the splice donor sites changed from "G" in liverworts, hornworts, and spikemoss to "A" in ferns and flowering plants (Fig. 6d and Supplementary Fig. 57). The copy of *VPS55* that does not have the microexon is derived from upstream intron retention in spikemoss due to the loss of splice donor site adjacent to the microexon. This indicates that the two *VPS55* gene copies evolved from a common ancestral gene with two alternative transcripts, and the two copies divergently evolved to either keep or lose the 1-nt microexon.

## Discussion

In this study, we identified >2000 small microexons (1–15 nt), of which >1000 were previously unannotated, in diverse genomes from flowering plants to algae. Moreover, we identified a sub-stantial fraction of annotated microexons that were not supported by transcriptome sequence reads. These results we believe will greatly improve plant genome annotation. In fact, for some current existing genome annotations, such as rice and maize, most microexons were missing, and most annotated microexons were not supported by RNA-seq junction reads. Because about half of plant microexons are not in multiples of three, such annotation errors will often result in frameshifted gene models and inaccurate peptide predictions, hindering inference of protein function. This is problematic because microexons are often found in gene models for proteins with fundamentally important cel-lular roles including transcriptional regulation, post-translational processing, metabolism, and intracellular movement. Because most coding microexons are included in major transcript iso-forms of their parent genes and conserved in plants, they are not likely to be functionally relevant as alternative exons.

When comparing the genome annotation tools with their performances in microexon and regular exon annotation in 10 plant species used in this study, we found that the HMM-based gene prediction program, Gnomon based on Genscan[29], devel-oped at NCBI is the most efficient one to detect microexons and regular exons, while Fgenesh and MAKER-P annotations are more likely to miss microexons. Regardless, standard genome annotators appear to be limited at identifying microexons in plant genomes. Therefore, using expression data such as RNA-seq and EST sequences with specifically designed tools, such as the pipeline we are reporting here, is necessary for microexon annotation in a given reference genome. The output format of our pipeline and microexon modeling is compatible with other gene annotation tools and could be easily incorporated.

To find the best tool for microexon identification, we com-pared the performance of two regular RNA-seq mapping tools, HISAT2 and STAR, and a de novo mapping tool, OLego using

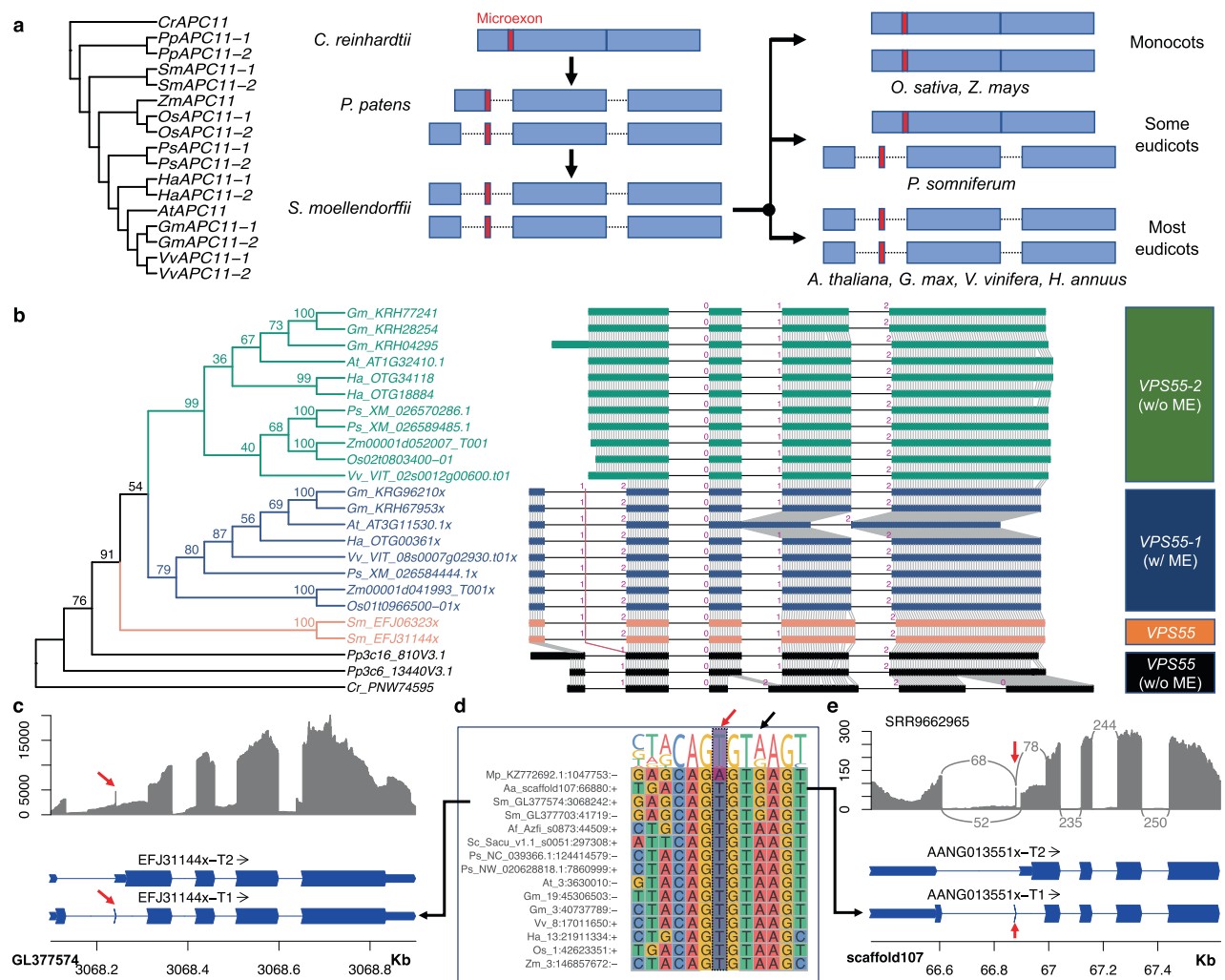

**Fig. 6 The evolution of two 1-nt microexons in *APC11* and *VPS55* from 10 plant species. a** Phylogenetic tree of *APC11* genes and the 1-nt microexon gain and loss. The transcript structures only contain coding regions. Boxes are coding exons and dotted lines are introns. **b** Phylogenetic tree, multiple sequence alignment, and gene structure comparison of *VPS55* genes. Only coding regions are shown in the block alignment. ME microexon, w with, w/o without. Transcripts end with an 'x' to distinguish from annotations. **c** RNA-seq read depth and two assembled transcript variants for one copy of *VPS55* in *Selaginella*: one has the 1-nt microexon and the other lost because of intron retention. The red arrows point to the 1-nt microexon. **d** Comparison of splice site sequences flanking the 1-nt microexons in *VPS55*. The highlighted purple box indicates the 1-nt microexon and the red arrow points to the position of the splice donor site, changed from "G" to "A", which causes the 1-nt microexon to gain a stronger splice site. The sequence labels refer to the position of the 1-nt microexon on each genome in the format: species_chromosome:position:strand. **e** RNA-seq read depth and two assembled transcript variants for the single copy *VPS55* in the hornwort *Anthoceros angustus*. Here, the RNA-splicing around the 1-nt microexon is complex. Arcs indicate the splicing junctions, and the numbers indicate the junction read counts. The red arrows indicate the 1-nt microexon. Please see Supplementary Fig. 57 for more species.

*Arabidopsis* and rice RNA-seq datasets (Fig. 1). According to this result, we developed and optimized an RNA-seq analysis pipeline to identify microexons in plants by integrating OLego and STAR, i.e., the splicing junctions discovered by OLego were used as additional splicing annotation for STAR to do short read mapping. Our pipeline, which interweaves STAR with OLego, largely improved the microexon identification in plants. Existing applications of OLego, such as FINDER[33], merely use OLego to map unmapped reads after STAR mapping. In addition, we also developed and validated an efficient computational method MEPmodeler[51] to predict microexon in plants, based on sequence homologies to a set of 45 well-conserved microexons. Because this method does not require any RNA-seq data, and thus it can be used to greatly expand microexon detection to genomes that lack sufficient RNA-seq data. Notably, the prediction method was able to efficiently identify very short microexons, such as 1-nt

microexon, which could not be identified by the RNA-seq-based method in some species. Thus, we believe that the development of these two approaches will be useful for improving genome annotations in plants.

The splicing of microexons has been studied in neurogenesis for animals[10,21]. In animals, the sizes of microexons are usually multiples of 3-nt and in-frame[9,10]. The regulation of inclusion or skip of microexons plays an important role in neuronal development. There is a highly conserved program of neuronal microexons which are mainly regulated by RNA-binding proteins like nSR100, RBFOX, and PTBP1[9,10]. Similar to in animals, splicing of microexons needs strong splicing signals, such as ISE. We found U-rich and T-rich motifs are enriched in introns around microexons in plants. Motifs with enriched T, i.e., U in RNA, are either the polypyrimidine tract (PPT), which are usually 15–20 base pair long[58], or UA-rich tract, which is required for

effective splicing of U2 introns in plants[59]. U-rich motifs are also binding sites of some RNA-binding proteins, such as TIA1/TIAL1 proteins and are adjacent to exons containing relatively lower ESE counts[60]. This U-rich region for microexon splicing has been experimentally validated in a 9-nt microexon in potato invertase[19]. The second type of motif, the G-rich motif, is a kind of ISE; G-rich sequence, usually containing at least one G-triplet, can recruit hnRNP H and F to enhance intron splicing[40]. Interestingly, unlike in animals that microexon skip frequently occurs, intron retention for microexons is more common in plants.

Microexons tend to rely on more splicing signals in the flanking introns, because microexons are too small to assemble the big spliceosome complex binding to both flanking introns simultaneously by the intron-definition model[61]. Therefore, there will be a competition of co-transcriptional splicing between two flanking introns, and the detained intron (unspliced intron) will be processed by PTS before the mature transcripts are released to the cytoplasm. Some splicing-related genes, such as PRMT5 and SKIP, were discovered to be associated with regular exon PTS splicing[25]. To test if these genes affect the splicing of microexons, we compared the intron unspliced ratios around microexons in total mRNA between mutants of these splicing-related genes and the wild type, and found that, although almost all these genes, such as SKIP and PP4R3A, had a certain level of associations with splicing of microexons, each of them had only a small effect, i.e., the unspliced ratios were slightly higher in mutants than in wild type (Supplementary Fig. 58). We also found that the Serrate-Associated Protein 1 (SEAP1)[62] was involved in the splicing of microexon flanking introns, especially downstream introns (Supplementary Fig. 58). This indicated that perhaps there are other unknown genes involved in PTS of microexons in plants.

Resolving the evolutionary history of green plants is important but challenging as some branches remain poorly resolved[63], despite intense investigations using various types of markers, such as conserved single copy genes[64], their transcripts[65], and coding genes in plastids[66]. After a comprehensive search of microexon-tags in 132 plant genomes, we hypothesized that microexon-tags may serve as useful markers in plant evolution studies, because microexon-tags contain gene structural information, have fixed lengths of nucleotide and amino acid sequences, and are conserved among land plants. Indeed, the phylogenetic tree constructed from microexons largely agrees with previous studies. Because a microexon-tag may have multiple copies in a gene family in a given genome, we tested two different approaches to construct phylogenetic trees using microexon-tags. For Method 1, we inferred a consensus microexon-tag sequence for each species in each cluster (for each site, only the highest frequency base was used). The consensus DNA sequences from all clusters were concatenated into a single large matrix, and missing sequences were treated as gaps (-). A maximum likelihood tree was constructed based on the concatenated matrix using IQ-TREE2[67]. For Method 2, we constructed a maximum likelihood tree for each cluster that included all microexon-tags from each cluster using IQ-TREE2. The species tree was inferred from all microexon-tag cluster trees using ASTRAL-Pro[68]. Notably, both trees are quite similar in that they recover monophyletic groups for eudicots, monocots, and magnoliids, and a sister relationship between angiosperms and gymnosperms (Supplementary Fig. 59). Moreover, almost all orders with multiple sampled taxa are also monophyletic. As more genomic sequences become available, microexon-tags could serve as potential gene markers for molecular evolutionary study. Here, we only focused on the smallest microexons (≤15-nt) and the most highly conserved. If the analysis was extended to larger and slightly less conserved microexons, it is possible that the phylogenetic tree would be improved further. Nevertheless, it is already clear that microexons exhibit

an evolutionary signal for given phylogenetic tree consistent with previous phylogenomic studies[64–66].

## Methods

**Plant species and RNA-seq datasets.** We selected 10 representative plant species representing a green alga, moss, spikemoss, and various flowering plants among monocots and dicots (Fig. 1c). Their reference genomes and gene annotations were obtained from Ensembl Plants (https://plants.ensembl.org) except for *Papaver somniferum*, whose reference genome and gene annotation information were obtained from NCBI Genome (https://www.ncbi.nlm.nih.gov/genome/). For rice (*O. sativa*), the MSU7 annotation was also used for comparison (http://rice.plantbiology.msu.edu). We carefully selected RNA-seq datasets from public databases and made samples as representative as possible to cover different tissues and developmental stages from various environmental conditions. For each species, the expression atlas was primarily collected, and the stress-induced expression profile was added if available. A total of 990 RNA-seq datasets were collected from NCBI SRA for all 10 plants (https://www.ncbi.nlm.nih.gov/sra), and with between 42 and 284 datasets for each individual species (Supplementary Data 1).

For the performance comparison of different mapping tools in microexon identification, two 50-bp and two 100-bp Illumina RNA-seq datasets were collected from *Arabidopsis* (SRA accessions: SRR3581695, SRR3581709, SRR14209167, and SRR14209168) and rice (SRA accessions: DRX000664, DRX000672, SRR5126147, and SRR5126148, RNA-seq datasets for microexon prediction validation included three 101-bp replicates (SRA accessions: SRR8434771, SRR8434772, and SRR8434773) from 4-day old roots of tomato (*Solanum lycopersicum*). RNA-seq datasets for the 1-nt microexon splicing in *VPS55* were listed in Supplementary Fig. 57. RNA-seq datasets for post-transcriptional splicing analysis were from NCBI project accession PRJNA591665 (only Illumine RNA-seq data being used) and the datasets for the mRNA splicing analysis in mutants of splicing-related genes were listed in Supplementary Fig. 58.

**Algorithm of microexon identification.** According to the performance comparison of three mapping tools (STAR[37], HISAT2[36], and OLego[32]), a pipeline integrating STAR and OLego was developed and optimized to identity microexons from short-read RNA-seq data. The pipeline is shown in Supplementary Fig. 1 and implemented in python. First, STAR genome indexes were generated from the reference genome fasta file and the GTF file of annotated transcripts. The clean RNA-seq fastq files were used to perform first pass mapping with STAR (parameters: --alignIntronMin 20 --alignIntronMax 20000 --outSAMtype None --outSJfilterReads Unique --outSJfilterCountUniqueMin 10 3 3 3 --outSJfilterCountTotalMin 10 3 3 3). Independently, OLego was used to map the same set of RNA-seq data to reference genome without transcript annotation provided (parameters: -e 3 -I 20000 --max-multi 5). The junctions were separately collected from the STAR first mapping and OLego mapping. These junctions were added to the STAR genome index for second pass mapping (parameters: --alignIntronMin 20 --alignIntronMax 20000 --limitBAMsortRAM 5000000000 --outSAMstrandField intronMotif --alignSJoverhangMin 20 --outSAMtype BAM SortedByCoordinate). Transcripts in each sample were assembled using StringTie with the reference annotation as a guide (parameters: -j 5 -c 5 -g 10 -G annotation.gtf) and merged together from all samples using transcript merge mode in StringTie (--merge), and the read coverage tables of introns, exons and transcripts (parameters: -e -B -G merged.gtf) were loaded into R package ballgown. For internal exon identification, only the exons with at least 5 junction reads on both sides of flanking introns in at least one sample were considered (Supplementary Fig. 1a). If alternative splicing occurs for one intron around the exon, the intron that has the most average junction reads in a population was used. Only introns with canonical splice sites (GT-AG or GC-AG) were considered in this study. The smallest microexons (1–15 nt) were identified in each species separately.

**RNA-seq simulation.** We used Polyester[69] to simulate RNA-seq reads from annotated microexons and corresponding transcripts from Araport11 in *Arabidopsis*. A total of 130 microexons with size ≤15 in 126 unique transcripts (one transcript with 5 microexons was excluded) were randomly divided into two equal groups: the true annotation group and the false annotation group. In the true annotation group, both simulated reads and GTF file were produced from the same transcripts. In other words, the GTF file correctly annotated the reads. On the other hand, the false annotation group was further randomly divided into two equal subgroups: the false positive subgroup, in which the GTF file contained the transcripts with microexons but simulated reads were produced from the transcripts with the microexon deletion, and the false-negative subgroup, in which the GTF file contained the transcripts with the microexons deleted but simulated reads were produced from the transcripts with microexons existing. For the simulation, two types of reads, 2 × 100 bp paired-end reads and 1 × 50 bp single-end reads were generated. For each type of reads, 10 replicates were generated by Polyester[69].

**Microexon splicing analysis.** To study the difference in the usage of ISEs in introns around microexons and regular long exons, we collected introns around constitutively spliced (CS) microexons in four plant species: *A. thaliana*, *G. max*, *O. sativa*, and *Z. mays*. As a control, we randomly selected 1000 CS introns, i.e., excluding the introns with any intron retention events. Here, we defined intron

retention as an unspliced intron located within a longer exon in assembled transcript from RNA-seq data. MEME was used to conduct differential enrichment mode analysis (http://meme-suite.org/tools/meme)[39]. We used sliding windows to analyze U-rich and G-rich motifs in intron sequences between 10-nt downstream of 5′ splice site to upstream 10-nt of 3′ splice site. For U-rich motif analysis, the window size is 20-nt and step size is 1-nt. A U-rich motif is counted if there are >80% T in a window. For G-rich motif analysis, the window size is 10-nt and step size is 1-nt. A G-rich motif is counted if there are >80% G in a window.

For detained intron (PTS intron) analysis, we used the same RNA-seq datasets from Jia et al.[25] and applied similar method as described previously[25]. In brief, for each flanking intron of an internal exon, the percentage of intron retention (PIR) was calculated as follows: $PIR = 100 \times (i_1 + i_2)/(i_1 + i_2 + 2e + 2o)$, where $i_1$ and $i_2$ are the numbers of junction reads spanning exon-intron and intron-exon junctions, respectively, $e$ is the number of splicing junction reads connecting to two adjacent flanking exons, and $o$ is the number of other alterative splicing junction reads. The junction reads were required to extend at least 4-nt from the intron boundary. The PIR value was set to missing (NA) if the number of total junction reads was <10. The Illumina RNA-seq datasets contained nine samples for polyadenylated RNA bound on chromatin (CB), in the nucleoplasm (NP), and in the cytoplasm (Cyto), with three biological replicates per sample[25]. Chromatin-bound polyadenylated RNAs are the input of the post-transcriptional splicing, and it is the initial time point of the post-transcriptional splicing process. Unspliced ratio of introns was estimated based on PIR value and only constitutively spliced introns (the average unspliced ratio < 0.2 in cytoplasm samples) were used in the analysis according to Jia et al.[25]. A detained intron requiring PTS was defined as unspliced ratio >0.1 in CB samples.

### Algorithm of microexon classification and modeling.

To study conserved microexons in plants, we focused on coding microexons. For transcripts obtained from our RNA-seq analysis pipeline, the function *getSeq* from the R package BSgenome was used to extract transcript sequences. The R package ORFik was used to get ORF locations on the reference genomes with the function *findMapORFs*. Only the longest ORF for each transcript was kept. CDS sequences were extracted from the reference genome according to the ORF positions, and the function *translate* from the Biostrings package was used to translate CDS sequences into protein sequences. The online tool GenomeNet was used to find protein motifs with default parameters (https://www.genome.jp/tools/motif/). If the peptide encoded by a microexon was located within or 5 amino acids (aa) away from a motif of a protein encoded by the parent transcript, we assigned that the microexon was involved in encoding that protein motif. If a microexon encoded more than one motif, only the most significant motif with the smallest *E*-value was chosen.

The pipeline of microexon classification and modeling is shown in Fig. 4a. First, we extracted 108-nt in-frame coding sequences around the microexon, called the microexon-tag (microexons in non-coding regions were not considered). More specifically, 108-nt continuous coding sequence, containing the microexon in the middle flanked by parts of adjacent exons, was extracted from the coding sequence of the longest ORF. To ensure the 108-nt sequence translated in-frame, it was shifted 1-nt forward (phase 1) or 1-nt backward (phase 2) or left unshifted (phase 0). If any side reached to the end of the ORF (start codon or stop codon), the length of sequence would be <108-nt but also in-frame. All extracted 108-nt sequences were translated into 36-aa peptide sequences and divided into groups according to their sizes and phases of microexons. For microexons with the same size and the same phase, the 36-aa peptide sequences were clustered using the *hclust* function in R stats package. A distance matrix was calculated based on the scores of pairwise alignments using *pairwiseAlignment* function with a BLOSUM62 substitution matrix in the R package Biostrings. A cluster was defined with a cut-off score of 50 and requiring a minimum of three species to have the microexon in the cluster. The most significant motif with smallest *E*-value in the cluster was picked to represent the motif encoded by the cluster. The unicellular algae, *C. reinhardtii*, was excluded for clustering and the following prediction because of low sequence similarity with the nine land plants.

To increase species and sequence representation in the clusters, we searched NCBI plant EST database with consensus sequence in each cluster using blastn (parameters: -word_size 7 -penalty -1 -reward 1 -ungapped -max_target_seqs 10000 -qcov_hsp_perc 80), and then extracted the corresponding 108-nt sequences matched to the cluster. The sequences having in-frame stop codons were removed. We also removed sequences shorter than the microexon-tags in the cluster. In each cluster, we merged all 108-nt EST sequences from the blastn search with the microexon-tags, removed redundant sequences and used the unique 108-nt coding sequences to construct a Position Weight Matrix (PWM) using the *PWM* function in the R package Biostrings. DNA and aa logos in each cluster were created based on the PWM using the R package ggseqlogo and ggplot2. To predict microexons in plants, we developed an R package, MEPmodeler[51], to model microexons in plant genomes independent of gene annotation and transcriptome data. According to the structure of microexon-tags in each cluster, the PWM can be divided into three or more parts: a microexon part and two or more flanking parts if any side of flanking sequence spanned two or more exons (e.g., 108-nt microexon-tags in Cluster 7 contains sequences from 5 exons). We searched each part of the PWM on the plus strand and minus strand of the whole genome using *matchPWM* function in R package Biostrings, allowing a gap between the adjacent parts (intron

presence) or no gap (intron absence). Gap sequences must contain canonical splice sites (GT or GC in the 5′ end and AG in the 3′ end). The gene structure comparison was based on multiple sequence alignment of protein sequences using *msaMuscle* function in the R package msa.

### Phylogenetic tree construction.

Plant genome sequences were downloaded from NCBI plant RefSeq genome database (March 2021) and some recently sequenced genomes were added. The microexon-tags in 45 clusters were predicted from each genome sequence, allowing no intron or any intron with size of 20 bp to 10 kb. Because our microexon-tags have the same length in the same cluster, one can immediately get accurate multiple sequence alignment for homologs. We used two methods to construct phylogenetic trees. For Method 1, we used the consensus string of microexons. The consensus DNA string was extracted from microexon-tags for each cluster in each species (for each site, only the highest frequency base was used). The consensus DNA strings from all clusters were concatenated end-to-end into a single large string. Missing strings were considered as gaps (-). The tree was constructed based on the concatenated large DNA strings of all species using IQ-TREE2 with 1000 ultrafast bootstrap replicates[67]. For Method 2, we constructed a gene phylogenetic tree for all microexon tags (108-nt sequences) in each cluster using IQ-TREE2 with 1000 ultrafast bootstrap replicates. The species tree was constructed based on all cluster trees using ASTRAL-Pro[68]. The phylogenetic trees were manipulated and visualized using R package ape and ggtree.

### Experimental validation.

Ten genes with microexons were selected from 10 microexons in *Arabidopsis*, soybean, maize and rice. The reference genomes *Arabidopsis* Columbia, soybean William 82, rice Nipponbare and maize B73 varieties were used. Total RNAs were extracted from *Arabidopsis* inflorescences, soybean seedlings, rice seedlings and maize roots using TRI Reagent (Molecular Research Center). And, cDNAs were synthesized using M-MLV Reverse Transcriptase (Promega). cDNAs were then amplified using Phusion™ High-Fidelity DNA Polymerase kit (Thermo Scientific) containing specific gene primers as shown in Supplementary Data 2. RT-PCR products were purified by GeneJET Gel Extraction Kit (Thermo Scientific), and then subjected to Sanger sequencing with specific sequencing primers listed on Supplementary Data 2.

### Reporting summary.

Further information on research design is available in the Nature Research Reporting Summary linked to this article.

## Data availability

All the RNA-seq data, genome annotations, and genome sequences used in this study were obtained from public databases and their accession numbers are listed in Methods and supplementary files. A total of 990 RNA-seq datasets were collected from NCBI SRA for 10 plants, and the accession numbers are available in Supplementary Data 1. For the performance comparison of different mapping tools in microexon identification, two 50-bp and two 100-bp Illumina RNA-seq datasets were collected from *Arabidopsis* (SRA accessions: SRR3581695, SRR3581709, SRR14209167, and SRR14209168) and rice (SRA accessions: DRX000664, DRX000672, SRR5126147, and SRR5126148). RNA-seq datasets for microexon modeling validation included three 101-bp (SRA accessions: SRR8434771, SRR8434772, and SRR8434773) were from 4-day old roots of tomato (*Solanum lycopersicum*). RNA-seq datasets for post-transcriptional splicing analysis were from NCBI project accession PRJNA591665 (only Illumine RNA-seq data being used). Source data are provided with this paper.

## Code availability

All codes for microexon discovery, microexon clustering, and microexon modeling generated in this study have been deposited in Zenodo (https://doi.org/10.5281/zenodo.5815987) and in Github (https://github.com/yuhuihui2011/MEPsuite)[70]. MEPmodeler, an R package for microexon modeling in plant genomes, has been deposited in Zenodo (https://doi.org/10.5281/zenodo.5816080) and in GitHub (https://github.com/yuhuihui2011/MEPmodeler)[51].

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

## Acknowledgements

We thank Dr. Jixian Zhai for sharing *Arabidopsis* PTS-intron list with us. We thank Hanh Nguyen and Thomas Clemente for providing Williams 82 soybean seedlings. We also thank all the research teams and individuals for sharing the genome sequences and RNA-seq datasets on public databases. The computations in the analysis of large-scale RNA-seq and genomic data were run on the bioinformatics computing platform of the National Key Laboratory of Crop Genetic Improvement, Huazhong Agricultural University. This project was supported by the National Science Foundation (Award #: OIA-1557417 to C.Z. and B.Y., and Award #: MCB-1818-82 to C.Z. and B.Y.) and the Nebraska Soybean Board (Award 20R-09-1/2 #1739 to C.Z.).

## Author contributions

H.Y. designed and performed research and developed algorithms. H.Y. and W.X. analyzed data. H.Y., B.Y., J.P.M., and C.Z. wrote the paper. M.L., B.Y., J.S., G.S., J.C.S, and H.W. conducted experiments and validation. C.Z., J.P.M., B.Y., and W.X. acquired funding and supervised the project.

## Competing interests

The authors declare no competing interests.
