## [Peer Review File · Nature Communications]

Pervasive misannotation of microexons that are evolutionarily conserved and crucial for gene function in plantsReviewers' Comments:

Reviewer #1:

Remarks to the Author:

Yu et al report a comprehensive analysis of microexons in plants. Microexons have been now studied relatively extensively in animals, but barely in plants. Yu et al implement a novel pipeline based on existing mapping tools (STAR and Olego) and identify dozens/hundreds of microexons in plants, convincingly showing that microexons are not well annotated in plant genomes (they are either missing or wrongly annotated without transcriptomic support). Then, they identify 45 clusters of microexons based on sequence length and similarity, and found that these are overall highly conserved across plants. Finally, the authors highlight two examples of 1-nt microexons (including a novel case).

While I appreciate the work and I personally found it interesting, I am afraid I am not convinced that the results are suitable for a broad readership and may fit better in a more specialized journal. Most of the results are related to genome annotation. This is certainly an important topic but a rather specialized one. On the other hand, the finding of 45 groups of conserved microexons is surprising, but their functional relevance as microexons is unclear. Detailed comments:

- 1) The authors introduce a new pipeline to identify microexons in plants. They identify multiple microexons in different species. However, the benchmarking is rather limited, since they only show how many microexons they identify with each splice mapper, favouring total numbers, but not a balance between true/false positives/negatives, which is not estimated. Also, they should definitely compare the results of their pipeline with those obtained running Microexonator (since this is a well-tested software that can work with any genome) and they could also compare their results with VAST-TOOLS for Arabidopsis. Finally, it is not clear what the distribution of inclusion levels for these microexons is (are they mainly constitutive or transcriptomic noise?).
- 2) Whereas Fig 1 shows hundreds of microexons, the claim from Fig 4 is that most microexon-mapping reads belong to the cluster of 45 conserved microexons. Does it mean the number of microexons in plants is rather limited to homologs of the 45 groups? This should be better clarified.
- 3) Most of the results relate to the quality of genome annotations in plants. They convincingly show that most genomes are not well annotated for microexons, but depending on point 2 this might be more or less relevant. Also, they do not provide an obvious solution for microexon annotation that genome annotation tools could easily incorporate and therefore the impact of these findings might be limited.
- 4) I got quite confused by their microexon "prediction". The authors define 45 clusters of microexon tags based on microexon sequence length and similarity, which correspond to 45 microexon homology groups, if I am not mistaken. Then, their "predictor" is really a search for homologs in other genomes based on sequence similarity and it cannot therefore be considered a proper de novo predictor, if I understood well. Is it the case? If so, they should re-word this section.
- 5) The conservation of these 45 clusters is remarkable. However, it was not straightforward to understand the associated results. In Fig 5, I ended up interpreting that a white cell means the microexon sequence could not be found and a red box that the sequence is there, but joint to a neighbouring exon through intron loss (assuming the microexon is ancestral). However, they write that the white boxes "indicates a missing data" (i.e. NA?), which is a bit confusing. Please use more precise wording.
- 6) It is not clear whether any of these microexons is alternatively spliced, which would make them more functionally exciting. If I understood well, they are "simply" tiny constitutive exons generated by two close intron insertions. While this is mechanistically interesting, it is functionally not very much so,

since it is a constitutive sequence of the protein whose skipping would often cause a frame shift. Under this scenario, the fact that these microexons are conserved is not so striking, since rates of intron gain and loss in land plants are known to be rather low (i.e. the microexons cannot be lost without negatively impacting the gene). Therefore, I think the word "crucial" in the title is misleading, particularly to those researchers in the alternative splicing field (or animal splicing field). In other words, the fact that a constitutive exon or microexon is important for gene function is expected.

7) Line 272: "the most common protein motifs encoded by microexons were" should better read "the most common protein motifs partly encoded by microexons were".

Reviewer #2:

Remarks to the Author:

In this study, Yu and co-authors performed bioinformatic analysis of microexons in plant. By improving bioinformatic pipelines, the authors improved annotation of microexons in 10 plant species based on public available RNA-seq data. More than 2000 microexons are identified including many that have not been noticed before. The authors then explored the possible splicing mechanism of microexons. The authors also developed an microexon prediction tool by establishing microexon-tags clusters. The authors then showed that such micro-exon tags are evolutionarily conserved and can be used in phylogenetic analysis. Finally, the evolution of two 1nt microexons are analyzed. Overall, this work is quite substantial in terms of microexon annotation and prediction, which should be useful to the plant research community. On the other hand, not all conclusions are supported by the data and some data are overinterpreted or misinterpreted. Additional works are needed before it reach publication standard. Below I summarize my concerns in detail.

1. Related to Figure 2, the authors concluded that the microexons in Arabidopsis are mainly spliced post-transcriptionally. There are several problems related to this conclusion as detailed below. Overall, the microexons are perhaps spliced somewhat slower than conventional exons (which is not surprising), but the data does not support they are predominantly spliced post-transcriptionally.

A. The author defined post-transcriptional splicing by looking at the polyadenylated fraction of chromatin-bound RNA, and an unspliced ratio >0.1 in this fraction was used as a standard for PTS. This standard is simply unsuitable for this study, for example, unspliced ratio of 0.1 means there are 10% of polyadenylated RNA on chromatin still harbors the intron next to the microexon. In another word, 90% of microexon is already spliced before polyadenylation. This actually means the majority of splicing events happened co-transcriptionally. In the previous paper (ref 25), a PTS intron means there is certain levels of splicing needed after polyadenylation at this intron, but this doesn't mean that the cutting out of the intron mainly occurs post-transcriptionally. In order to conclude that microexons are mainly spliced co or post-transcriptionally, a more stringent standard should be used when defining PTS. For example, how many of the microexons have an unspliced ratio of more than 0.5 as the case showed in Fig. 2e.

B. It is unclear to this reviewer why the author only looked the polyadenylated fraction of chromatin-bound RNA when defining PTS. In this way, the information of co-transcriptional splicing is missed. The author should look at the long reads sequencing data of CB-RNA in which the information of both polyadenylated and nascent RNA are kept. In this way, the author could accurately estimate to what extent these microexons are spliced before and after polyadenylation at the chromatin level.

C. Outside of plant field, the word post-transcriptional splicing means the splicing off the chromatin instead of on chromatin. It is unclear to this reviewer why the author want to specifically highlight the PTS on chromatin and what would be the biological implication for such a phenomenon. Splicing after polyadenylation but on chromatin means there is no strict order between polyadenylation and splicing on chromatin but does not necessarily mean such splicing is uncoupled with transcription or Pol II. A

more accurate definition of PTS is required here to avoid confusions.

D. Line 255 to 258, the author found the introns before and after the microexons are often removed consecutively rather than simultaneously. This has nothing to do with co or post transcriptional splicing. In co-transcriptional splicing, the introns removal in general follows "first come, first served" rule (ref 25 to 27) but this is just a general trend while not a strict rule. For example, exons flanked by weak splicing site might take longer time to be splice during transcription. The time that an intron sit on chromatin plus the strength of splice site determines how efficient such an intron is being removed on chromatin(or co-transcriptionally). The author misinterpreted the data here.

E. There is actually high-quality Pol II NET-seq data available for Arabidopsis (J. Zhu et al, Nature Plants). Would Pol II pause less on microexons comparing with conventional exons if microexons are spliced slower ?

2. Related to Figure 4 and Figure 5, it is nice that the authors developed the prediction tool based on micro-exon tags. These tags are based on clustering of sequences with homology, so it is not surprising they harbors evolutionary information. But why this could be preferred comparing with other markers that commonly being used in phylogenetic analysis ?

3. It is interesting that only 67 microexons are constitutively spliced in Arabidopsis. How often the microexons are alternatively spliced ? If they often subject to alternative splicing, what's the relative abundance of transcripts that harbors microexons comparing with those without microexons ? Are those alternative splicing events regulated or related to stress or development ? These analysis would provide insights about how important of those microexons in plant.

4. In Figure 6, the authors looked the evolution of two 1nt microexons in APC11 and VPS55. The underlying biological significance of such microexon is missed. Do they cause any consequence on the function of proteins or on RNA expression ?

Minor point:

1. The labels of species on Figure 1C and Figure 5 are difficult to read when print out.
2. Figure 6a, the meaning of different colors is unclear. Please also label where is the ATG on those schematics.

Reviewer #3:

Remarks to the Author:

1. What are the noteworthy results?

The authors identified and analyzed numerous microexons ranging from 1 to 15bp that were previously unannotated, but may be essential to vital functionality in many plant species. They claim that these portions of genomes are not annotated or rather "pervasively misannotated," as well as describe genetic signals indicating their existence, develop a computational tool for analyzing RNA-Seq data to determine their existence, and apply the tool to a set of genomic data from several plant species.

2. Will the work be of significance to the field and related fields? How does it compare to the established literature? If the work is not original, please provide relevant references.

The work in this paper focuses on microexons that are of the smallest magnitude in terms of length in number of nucleotides. Furthermore, the focus of the manuscript is on plant genomes that have been less studied in terms of the discovery of the microexons. The work in the paper is significant to other fields that seek to identify and examine these microexons in different species, especially at smaller sizes, and can also inform research into specific plant species and their proteins.

3. Does the work support the conclusions and claims, or is additional evidence needed?

The author's claim of pervasive misannotation among extant plant genetic annotations is slightly too broad; after all, the existing microexons aren't necessarily misannotated as something else, they are simply not annotated as they have not been discovered. I think misannotation could effectively be replaced with a different word, and I also believe that there should be more focus on providing evidence for this claim if this is the most critical part of the paper in the author's eyes. Their more profound findings in my opinion are the simple abundance of microexons and the technique they provide to discover these. Unfortunately the methods section of the draft does not really give a clear outline of the overall novel algorithm that is used for this identification; instead the authors opt to describe textually the many different existing software programs that they chained together. In its current form I would suggest that the paper be revised to include a clearer outline of the method that is being proposed by the authors with a potential flow diagram.

Figure 1b shows a comparison of aligners and their combined uses in terms of the number of reads aligned to at least three exons with a middle exon as a microexon. It is possible that a lot of such reads, especially those aligned to novel microexons, may be incorrectly aligned, which can lead to misinterpretation. I suggest that the authors randomly choose, perhaps 100 such cases (10 cases per species) and perform RT-PCR to calculate false positive rates.

4. Are there any flaws in the data analysis, interpretation and conclusions? Do these prohibit publication or require revision?

Again, the methodology seems to be a pipeline of existing bioinformatics tools and statistical tools from various R software packages (i.e., Ballgown, ORFik, etc...). It would be very helpful to the readers if there was a flow diagram clearly depicting the flow of the data from RNA-Seq through the various portions of the proposed pipeline with labeled intermediaries. As it stands, the methodology likely meets the standards in the field, which often include procedures that can be expressed as a conglomeration of various extant packages. However, it is difficult to read the methodology section in this manner, and the methodology is one of the most critical contributions of this manuscript in my opinion, even though the authors suggest that their resulting findings that microexons are pervasively missannotated is more critical to the manuscript.

5. Is the methodology sound? Does the work meet the expected standards in your field?

The authors developed two methods, one being a pipeline for identifying microexons using RNA-seq data and the other, MEP, being a program for predicting microexons using plant genomes. Only the latter program's code is provided on GitHub. Though the authors provide enough detail to reproduce the work that was generated by the former pipeline, I think providing some scripts with some example sequencing data would be useful so that other researchers could quickly and efficiently annotate other plant genomes processed through the pipeline.

Overall comments: The paper is interesting and relevant, and there is a good set of results and graphics displaying those results. The methodology section leaves something to be desired, however. It would be very nice for a clear separation of the new methodologies introduced by the authors (the prediction tool) and the pipeline constructed with other previously created technologies. To aid with this, I suggest the inclusion of a diagram showing the flow of the RNA-Seq data with intermediary data labeled. In its current form, the paper seems to focus more on the results than on the methodology applied to produce these. I believe that the focus here should be more on the methodology and that the focus on misannotation of microexons in plants should be more appendant to the main findings here, which are critically the methods of discovering very small exons in RNA-Seq data.

One minor point is that there are two Supplementary Table 1s, which is confusing. The first instance of Supplementary Table 1 is almost identical to Table 1, so it can be removed.

Reviewer #4:

Remarks to the Author:

The manuscript by Yu et al. describes a pipeline to accurately detect micro-exons (<51 nt) to improve gene annotation. They have further explored the various characteristics of micro-exons and have performed statistical tests to show how those features differ from other exonic features. The implementation of the pipeline, as described in the manuscript, is well organized, clearly presented with useful explanations, figures, and provides useful data for the community. It is particularly valuable for researchers who wish to focus on microexons and its impact on various genome regulatory processes and plant evolution. The approach is comprehensive and the article is a very interesting read. The validations carried out by the authors make their claim even more convincing.

It would be great if the authors could consider the following questions and comments:

The authors have mentioned that they use relaxed thresholds for aligning reads using STAR and OLego (line 185). In the methods section, they have included the parameters. For the first pass of STAR, they use (`--outSJfilterReads Unique --outSJfilterCountUniqueMin 10 3 3 3 --outSJfilterCountTotalMin 10 3 3 3`) as parameters. The default parameter setting is (`--outSJfilterReads All --outSJfilterCountUniqueMin 3 1 1 1 --outSJfilterCountTotalMin 3 1 1 1`). In the second round of STAR, the parameters are set to (`--alignSJoverhangMin 20`) and the default for this parameter is 5. In both the rounds, it seems like the mapping parameters were more stringent than the default, but the authors claim that they used relaxed parameter settings. Could you please help us understand this difference and what made the thresholds more relaxed?

A recently published software, called FINDER, implements STAR and OLego to perform alignment precisely to detect micro-exons and perform gene annotation (<https://bmcbioinformatics.biomedcentral.com/articles/10.1186/s12859-021-04120-9>). Even though the FINDER paper does not explore the characteristics of micro-exons in great depth, it seems to be the one of the first softwares to use OLego to detect micro-exons and therefore warrants a brief discussion in that section of the manuscript.

(Comment only) Several plant species have poor gene annotation. I am curious to see how the pipeline will behave if STAR was executed without any support of existing annotations.

What were the reasons that only microexons with canonical splice sites were considered. Would there be a possible method to detect transcripts with internal micro-exons that do not have canonical splice sites?

The authors have used Stringtie to assemble the alignments into transcripts. Then they have used read coverages of introns, exons, and transcripts. Stringtie can be executed to generate coverages for transcripts, but does it also generate read coverages for introns and exons? In addition to that, it will be nice to provide the parameters used to execute Stringtie.

It is great that the authors conducted validation tests with a subset of transcripts. RT-PCR tests can pull out the matured mRNA which is devoid of introns. Could you explain how these transcripts were mapped to the reference? From figure 4c, I understand that the red-colored alignment is different from the blue-colored alignment. How did you confirm that the blue-colored alignment (with the 1 nt micro-exon) is the correct alignment (and not the red-colored one), from RT-PCR, Sanger Sequencing, and subsequent mapping? A similar example can be seen in figure 6a, where the red-colored portion in the *P.patens* transcript is a micro exon. In the first example, it is a part of the first exon, but in the second example it is an exon all by itself. I wonder how you would validate it using sequencing. Clarification is appreciated.

Is there a software tool available that implements the algorithm discussed that can readily detect

micro-exons when provided with the reference and RNA-Seq data?

In Figure 1a, could the author define the 'M' and 'N' symbols used in the figure for readers unfamiliar with CIGAR notation.

For the explanation of figure 2b, a small note on ISE-hex3-D motif would help to understand the third x-axis label better.

The caption for Figure 3b is a bit confusing and hard to read. Rewording is recommended.

The word RNA-Seq is inconsistently used in the manuscript ('RNAseq', 'RNA-Seq', and 'RNA-seq').

Line 718, the sentence ends by referencing a superscript.

Response to Reviewer #1's Comments to the Author

Yu et al report a comprehensive analysis of microexons in plants. Microexons have been now studied relatively extensively in animals, but barely in plants. Yu et al implement a novel pipeline based on existing mapping tools (STAR and Olego) and identify dozens/hundreds of microexons in plants, convincingly showing that microexons are not well annotated in plant genomes (they are either missing or wrongly annotated without transcriptomic support). Then, they identify 45 clusters of microexons based on sequence length and similarity, and found that these are overall highly conserved across plants. Finally, the authors highlight two examples of 1-nt microexons (including a novel case).

While I appreciate the work and I personally found it interesting, I am afraid I am not convinced that the results are suitable for a broad readership and may fit better in a more specialized journal. Most of the results are related to genome annotation. This is certainly an important topic but a rather specialized one. On the other hand, the finding of 45 groups of conserved microexons is surprising, but their functional relevance as microexons is unclear. Detailed comments:

Response: Thanks for the comments. We agree that a large portion of the manuscript is devoted to the pervasive misannotation of microexons in plants, but this is likely an issue that affects annotations of many eukaryotes. We would also like to emphasize that, besides the genome annotation method, we also describe the structure and splicing process, and we have made substantial revisions to the manuscript to more clearly convey the functional and evolutionary importance of microexons in plants. The proteins encoded by genes with microexons are related to development, stress response, metabolism, and more. Due to the extreme short size and unique splicing mechanism of microexons, the microexons are evolutionally conserved and are usually included in primary transcripts, although alternatively spliced transcript isoforms, that either lack the microexons or that exhibit intron retention, also exist. Therefore, this study should have a broad readership.

1) The authors introduce a new pipeline to identify microexons in plants. They identify multiple microexons in different species. However, the benchmarking is rather limited, since they only show how many microexons they identify with each splice mapper, favouring total numbers, but not a balance between true/false positives/negatives, which is not estimated. Also, they should definitely compare the results of their pipeline with those obtained running MicroExonator (since this is a well-tested software that can work with any genome) and they could also compare their results with VAST-TOOLS for Arabidopsis. Finally, it is not clear what the distribution of inclusion levels for these microexons is (are they mainly constitutive or transcriptomic noise?).

Response: Thanks for the insightful suggestion! Based on the comments, we conducted extensive analyses and comparisons with other tools. These results were included in the revised manuscript, specifically in the first section of Results (Lines 144-155) and supplementary Fig. 2. For comparisons, both real RNA-seq data and simulated data were used. The simulated data were used to estimate true and false positive rates of microexon discovery. These analyses showed that VAST-TOOL, MicroExonator, and our method all have low false discovery rate, and our method has better true positive rate. After applying all tools to the real RNA-seq data, we showed that our tool could find more microexon candidates in plants and could confirm more of the reference annotated microexons. Actually, before submitting the manuscript, we had tested MicroExonator and communicated with MicroExonator's author, Dr. Guillermo E. Parada, for performance evaluation. Guillermo said they "designed and optimized

MicroExonator to work with ~100 nt long reads". We have a lot of RNA-seq samples with ~50 nt short reads and the discovery of MicroExonator would not be very productive. Our method combining OLego and STAR can handle short reads such as ~ 50 nt, although long reads are better.

To estimate the inclusion levels of microexons, we calculated PSI (percent spliced in) of each microexon as the percentage of transcript isoforms that contain the microexon in the parent genes. The distribution of PSIs in 10 different plants showed that coding microexons were usually included in major transcript isoforms, suggesting they are functionally important. For the minor isoforms that lack the microexons, we cannot say whether they are also functionally important or represent transcriptomic noise. We added these details in the main text and supplementary Fig. 3 in the revised manuscript.

2) Whereas Fig 1 shows hundreds of microexons, the claim from Fig 4 is that most microexon-mapping reads belong to the cluster of 45 conserved microexons. Does it mean the number of microexons in plants is rather limited to homologs of the 45 groups? This should be better clarified.

Response: According to Fig. 4e, most microexon-spanning reads belong to these 45 microexon clusters because these microexons tend to have high expression levels among all microexons. The high expression of these 45 modeled microexons provides ample data to allow them to be modeled in diverse species, and also increases the reliability that they are modeled correctly. The 45 microexon clusters cover about 70% of all coding microexons in plants (e.g., 70 of 106 microexons in *Arabidopsis*, 147 of 235 in soybean, 66 of 84 in rice, and 81 of 126 in maize). Although the 45 microexon groups only represent about 70% of coding microexons, the generally high expression of these microexons means that 90% or more of all microexon-mapping transcript reads will associate with these 45 microexons. That being said, there are quite a few additional microexons in plants, but they tend to be less conserved among species (~30% of microexons can be species specific) and often exhibit lower expression levels. In summary, there are more microexons besides homologs of the 45 groups. We added some words to explain this (Lines 367-369).

3) Most of the results relate to the quality of genome annotations in plants. They convincingly show that most genomes are not well annotated for microexons, but depending on point 2 this might be more or less relevant. Also, they do not provide an obvious solution for microexon annotation that genome annotation tools could easily incorporate and therefore the impact of these findings might be limited.

Response: For microexon annotation in plant genomes, we have two different methods: one is based on the RNA-seq read mapping, and the other is genome-wide prediction based on the microexon sequence and gene structures (now we call it MEPmodeler). For the revised manuscript, we released all codes for microexon discovery and prediction to GitHub (<https://github.com/yuhuihui2011/MEPsuite> and <https://github.com/yuhuihui2011/MEPmodeler>). For microexon annotation from RNA-seq data, users can use the *golden_map* pipeline in MEPsuite to get microexon annotation in GTF format. For microexon annotation without RNA-seq data, i.e. model prediction, users can use MEPmodeler to get microexon annotation, including genomic coordinates and peptide sequences. The outputs of these tools are compatible with other gene annotation tools, such as Gnomon, and could be easily incorporated

into the genome annotation pipelines. In the revised manuscript, we mention this compatibility with genome annotators in the discussion (Lines 491-495).

4) I got quite confused by their microexon "prediction". The authors define 45 clusters of microexon tags based on microexon sequence length and similarity, which correspond to 45 microexon homology groups, if I am not mistaken. Then, their "predictor" is really a search for homologs in other genomes based on sequence similarity and it cannot therefore be considered a proper de novo predictor, if I understood well. Is it the case? If so, they should re-word this section.

Response: We are sorry that the prediction method was not clearly described in the previous version of the manuscript. To clarify, the prediction method (which we have renamed MEPmodeler) does not merely search for homologs, like a blast. In fact, a homology search approach could not succeed in microexon modeling due to long intron gaps and short microexon sequences. It is true that we do a blast search, but this search is done to against the NCBI EST database to help us define a position weight matrix for each ME tag. This PWM is then used as part of the larger MEPmodeler workflow to model microexons from whole-genome data. During this process, besides the sequence similarity of both DNA sequences and peptide sequences, our MEPmodeler must also consider the structure of the microexon-tag, including the sizes, phases, and positions of microexons, the flanking introns and exons, and also the translated peptides (e.g., no stop-codon). These extra structural considerations are essential to the identification of short microexons from whole genome data. We re-worded this section for clarification (Lines 327-336).

5) The conservation of these 45 clusters is remarkable. However, it was not straightforward to understand the associated results. In Fig 5, I ended up interpreting that a white cell means the microexon sequence could not be found and a red box that the sequence is there, but joint to a neighbouring exon through intron loss (assuming the microexon is ancestral). However, they write that the white boxes "indicates a missing data" (i.e. NA?), which is a bit confusing. Please use more precise wording.

Response: We thank the reviewer for pointing out this issue! Yes, the white boxes indicate microexon-tag (not merely the sequence) could not be found. In the revised manuscript, we changed it to "White cells indicate microexon-tag could not be found (NA)" in the Fig. 5 legend. Yes, the red box means the sequences of microexons and flanking exons are there, but microexons are joined to one or two flanking exons. A red box could be either ancestral microexon that lacks one or two flanking introns, or descendant of a microexon that loses the flanking introns by reverse transcriptase (RT)-mediated intron loss. We used an example of APC11 for microexon gain and loss.

6) It is not clear whether any of these microexons is alternatively spliced, which would make them more functionally exciting. If I understood well, they are "simply" tiny constitutive exons generated by two close intron insertions. While this is mechanistically interesting, it is functionally not very much so, since it is a constitutive sequence of the protein whose skipping would often cause a frame shift. Under this scenario, the fact that these microexons are conserved is not so striking, since rates of intron gain and loss in land plants are known to be rather low (i.e. the microexons cannot be lost without negatively impacting the gene). Therefore, I think the word "crucial" in the title is misleading, particularly to those researchers in the alternative splicing field (or animal splicing field). In other words, the fact that a constitutive exon or microexon is important for gene function is expected.

Response: We are sorry for any confusion here. As we mentioned in our response to Question (1), we estimated the inclusion levels of microexons by calculating their PSIs, and the result showed that most PSIs of the coding microexons are in the range of 0.5-1. This indicates there is alternative splicing for microexons, but these microexons were usually included in the major transcript isoforms. We added this result to the main text and Supplementary Fig. 3 in the revised manuscript. Actually, microexons in plants do have a higher rate of alternative splicing, especially for intron retention. Because most of microexons are included in the major transcript isoforms ($PSI > 0.5$), the misannotation of these microexons may result in truncated proteins and incorrect annotation of protein function.

7) Line 272: "the most common protein motifs encoded by microexons were" should better read "the most common protein motifs partly encoded by microexons were".

Response: Thanks for the suggestion. We modified this sentence in the revised manuscript accordingly (Lines 223).

Response to Reviewer #2's Comments to the Author

In this study, Yu and co-authors performed bioinformatic analysis of microexons in plant. By improving bioinformatic pipelines, the authors improved annotation of microexons in 10 plant species based on public available RNA-seq data. More than 2000 microexons are identified including many that have not been noticed before. The authors then explored the possible splicing mechanism of microexons. The authors also developed an microexon prediction tool by establishing microexon-tags clusters. The authors then showed that such micro-exon tags are evolutionarily conserved and can be used in phylogenetic analysis. Finally, the evolution of two 1nt microexons are analyzed. Overall, this work is quite substantial in terms of microexon annotation and prediction, which should be useful to the plant research community. On the other hand, not all conclusions are supported by the data and some data are overinterpreted or misinterpreted. Additional works are needed before it reach publication standard. Below I summarize my concerns in detail.

Response: Thanks for the nice and constructive comments.

1. Related to Figure 2, the authors concluded that the microexons in Arabidopsis are mainly spliced post-transcriptionally. There are several problems related to this conclusion as detailed below. Overall, the microexons are perhaps spliced somewhat slower than conventional exons (which is not surprising), but the data does not support they are predominantly spliced post-transcriptionally.

Response: We are sorry for confusion by using misleading words. We did not intend to suggest that the majority (>50%) of flanking introns of a microexon could not be spliced out by co-transcriptional splicing and would need to be spliced out by post-transcriptional splicing (PTS). The phenomenon we wanted to describe (based on results in Fig. 2d) is that the flanking introns of many microexons tend to be more delayed in splicing (evidenced by an increase in the unspliced ratio, but not necessarily > 50%), which may be spliced by PTS. This phenomenon is in contrast to regular exons – for most regular exons, flanking introns are completely spliced by co-transcriptional splicing and do not need PTS. Yes, we actually quantitatively showed evidence that microexons are spliced slower than conventional exons. To avoid misunderstanding, we revised this part to better convey our meaning and we now use the term

“detained introns” that was previously used in animals and humans (Boutz et al., 2014, *Genes Dev*) in the revised manuscript.

A. The author defined post-transcriptional splicing by looking at the polyadenylated fraction of chromatin-bound RNA, and an unspliced ratio >0.1 in this fraction was used as a standard for PTS. This standard is simply unsuitable for this study, for example, unspliced ratio of 0.1 means there are 10% of polyadenylated RNA on chromatin still harbors the intron next to the microexon. In another word, 90% of microexon is already spliced before polyadenylation. This actually means the majority of splicing events happened co-transcriptionally. In the previous paper (ref 25), a PTS intron means there is certain levels of splicing needed after polyadenylation at this intron, but this doesn't mean that the cutting out of the intron mainly occurs post-transcriptionally. In order to conclude that microexons are mainly spliced co or post-transcriptionally, a more stringent standard should be used when defining PTS. For example, how many of the microexons have an unspliced ratio of more than 0.5 as the case showed in Fig. 2e.

Response: We are sorry that we could not clarify our statement. For microexons, we wanted to state that majority of microexons have a certain level of detained flanking introns (the unspliced ratio is not necessarily $> 50\%$), which require increased post-transcriptional splicing (PTS). We did not want to claim that the cutting out of flanking introns of microexons mainly occurs post-transcriptionally. Yes, maybe, many copies of flanking introns of microexons were already spliced before polyadenylation, but we found that many microexons still have a fraction of detained introns after polyadenylation. This phenomenon is in contrast to conventional exons - flanking introns of most regular exons were completely spliced by co-transcriptional splicing. We used the same criterion (unspliced ratio >0.1) as defined by the previous paper (Ref. 25) to define PTS-intron (or detained intron) and the author of Ref. 25 kindly shared their regular exons' PTS-intron list with us. We have checked unspliced ratios of all microexon-flanking introns, and there are very few microexons with an unspliced ratio of more than 0.5 as the case showed in Fig. 2e. Please see Fig. 2d, most of microexons have an unspliced ratio between 0.1-0.5 in CB samples. To avoid misunderstanding, we revised this part accordingly.

B. It is unclear to this reviewer why the author only looked the polyadenylated fraction of chromatin-bound RNA when defining PTS. In this way, the information of co-transcriptional splicing is missed. The author should look at the long reads sequencing data of CB-RNA in which the information of both polyadenylated and nascent RNA are kept. In this way, the author could accurately estimate to what extent these microexons are spliced before and after polyadenylation at the chromatin level.

Response: Thanks to reviewer #2 for this insightful suggestion. We tried using long reads from CB-RNA as suggested, but unfortunately long reads have high sequence errors and could not be used to accurately identify microexons. We have tried different long read correction methods and mapping methods, but could not solve the problem. Importantly, the information of splicing before and after polyadenylation at the chromatin level does not change the main conclusion of this analysis that microexon-flanking introns exhibit an increase of delayed splicing that likely requires PTS.

C. Outside of plant field, the word post-transcriptional splicing means the splicing off the chromatin instead of on chromatin. It is unclear to this reviewer why the author want to specifically highlight the PTS on chromatin and what would be the biological implication for such a phenomenon. Splicing after polyadenylation but on chromatin means there is no strict order between polyadenylation and splicing on chromatin but does not necessarily mean such splicing

is uncoupled with transcription or Pol II. A more accurate definition of PTS is required here to avoid confusions.

Response: Thanks to reviewer #2 for pointing out this key issue. Yes, we agree that the post-transcriptional splicing refers to the splicing process from dropping off the chromatin to releasing into cytoplasm. Technically, we cannot study the dynamics of the intron splicing during post-transcriptional splicing – extracting RNAs in the nucleus can lead to a mixture of all stages of intron splicing. However, because chromatin-bound polyadenylated RNAs are the input of the post-transcriptional splicing, we can use these data as the time-0 point of the post-transcriptional splicing process. Subsequently, all detained introns on chromatin-bound polyadenylated RNAs will be either spliced out by post-transcriptional splicing or permanently retained and sent to the cytoplasm. Therefore, the fraction of detained introns that get spliced out by PTS can be obtained by calculating the difference between the chromatin-bound polyadenylated RNA and mature RNA in the cytoplasm. This kind of study was originally designed by the authors of ref. 25, and we used their data to analyze microexons in our study. We added some words to explain this topic in the revised manuscript (Lines 96-99).

D. Line 255 to 258, the author found the introns before and after the microexons are often removed consecutively rather than simultaneously. This has nothing to do with co or post transcriptional splicing. In co-transcriptional splicing, the introns removal in general follows “first come, first served” rule (ref 25 to 27) but this is just a general trend while not a strict rule. For example, exons flanked by weak splicing site might take longer time to be splice during transcription. The time that an intron sit on chromatin plus the strength of splice site determines how efficient such an intron is being removed on chromatin (or co-transcriptionally). The author misinterpreted the data here.

Response: We are sorry for the confusion here. Yes, we totally agree with reviewer #2 about the comments on intron-splicing regulation. Yes, the time that an intron sits on chromatin affects the splicing, but here, we have to consider the status of a polyadenylated RNA bound on chromatin is a time point for our analysis (Ref.25 adopted the same assumption). Because we used data for chromatin-bound polyadenylated RNAs (from Ref. 25), we can see a snapshot of the intron-splicing status at a specific critical time-point – the end time point of co-transcriptional splicing process and the beginning of the post-transcriptional splicing process. As we mentioned in our response to Question (C), a chromatin-bound polyadenylated RNA is the time-0 of the post-transcriptional splicing process. Because the data show a snapshot of the intron-splicing status at a specific time point, any unspliced introns at this time point will have a later splicing than introns already spliced at this time point. We reported our observation on microexon introns based on this logic, which was originally employed in Ref. 25.

E. There is actually high-quality Pol II NET-seq data available for Arabidopsis (J. Zhu et al, Nature Plants). Would Pol II pause less on microexons comparing with conventional exons if microexons are spliced slower ?

Response: Thanks to reviewer #2 for providing this information and suggestion. We tried to do the analysis with the Pol II NET-seq data, accordingly. Unfortunately, although the sequence depth of the pNET-seq data is high, the coverage for a particular exon is very low (the median count of reads within the 50 bp away from the splicing site is only 2-5). We also could not compare microexons with regular exons far from them because, as reported in the paper (J. Zhu et al, Nature Plants), Pol II tends to accumulate near the transcription start site (TSS) and causes a bias toward increased coverage for exons close to the TSS. In fact, we detected some

evidence for differences in coverage for the upstream exon relative to the microexon and downstream exon, but given the potential bias in positional coverage, we cannot make firm conclusions about the technical vs. biological source of this result (Please see the following figure, Fig. 1, for the result). Therefore, we decided not to include this analysis in the revised manuscript.

Fig. 1: The accumulation reads in 3'end of microexons and the flanking exons using pNET-seq data. **a**, Total Pol II (pNET_total). **b**, Pol II with a Ser5P carboxyl-terminal domain (CTD) (pNET_Ser5P). **c**, Pol II with a Ser2P CTD (pNET_Ser2P). **d**, Pol II with an unphosphorylated CTD (pNET_unph). Two replicates were merged together. The raw sequence data were obtained from Gene Expression Omnibus (GSE109974) and were processed as described by Zhu et al., (2018). For each exon, reads within 50 bp away from the 3'end (100 bp region) were counted and log2 transformed, i.e., $\log_2(\text{count}+1)$. We also tested the 100-bp region centered with 5'end or median point of each exon, and the results were similar.

2. Related to Figure 4 and Figure 5, it is nice that the authors developed the prediction tool based on micro-exon tags. These tags are based on clustering of sequences with homology, so it is not surprising they harbor evolutionary information. But why this could be preferred comparing with other markers that commonly being used in phylogenetic analysis?

Response: Thanks to reviewer #2 for this excellent question. That is why we introduced the term “microexon-tag”, instead of using “microexon” directly. To model microexons, we predicted microexon-tags, which contain gene structural information, including microexon length and phase, have fixed lengths of nucleotide and amino acid sequences, and are conserved among land plants. In other words, microexon-tags include multiple types of information, which are all related to evolution and hence make microexon-tags a good marker for evolutionary study. Importantly, in phylogenetic analysis, the most challenging process is to find homologies among different species, and particularly to distinguish orthologs from paralogs, especially among species with distant relationships. Our microexon-tags, which must have the same length in the same cluster and must share additional structural information (flanking introns and exons), provide an additional level of information beyond sequence similarity which we expect will help identify true orthologs. More generally, this increased information enables an immediately accurate multiple sequence alignment for homologs, essentially avoiding errors during sequence alignment that can affect the assignment of sequence homology among species.

3. It is interesting that only 67 microexons are constitutively spliced in Arabidopsis. How often the microexons are alternatively spliced? If they often subject to alternative splicing, what's the relative abundance of transcripts that harbors microexons comparing with those without microexons ? Are those alternative splicing events regulated or related to stress or development ? These analysis would provide insights about how important of those microexons in plant.

Response: We thank reviewer #2 for the suggestion. By following reviewer #2's suggestion, we studied the splicing levels of microexons by calculating the distributions of PSI (percent spliced in). The result showed that most coding microexons are located in the major transcript isoforms. Our analysis based on regular RNA-seq data indicated that microexons do have higher rate of alternative splicing, especially for intron retention (Please see Fig. 2c). However, one needs to note that the intron retention could be because many microexons have detained flanking introns that require PTS. We added the result to the main text and supplementary Fig. 3 in the revised manuscript.

4. In Figure 6, the authors looked the evolution of two 1nt microexons in APC11 and VPS55. The underlying biological significance of such microexon is missed. Do they cause any consequence on the function of proteins or on RNA expression?

Response: We are sorry for any confusion here. Because about half of microexons (including these two 1nt microexons) are not in multiples of 3, the inclusion or exclusion of these microexons in the mature transcript will alter the translation of all downstream codons, which would have major effects on the protein sequence, structure, and function. Actually, we have described the significance and importance of the 1-nt microexon in the section "Two specific cases of 1-nt microexons APC11 and VPS55". In the manuscript, we stated that "Annotations lacking these 1-nt microexons predict proteins with shorter or different N-terminal peptides. For example, in flowering plants, the 1-nt microexon is usually not annotated in its host VPS55 gene, which results in the absence of N-terminal amino acids (13 amino acids in Arabidopsis)." To enhance this point, we added more words to describe the change of protein structure and function in the revised manuscript (Lines 425-428). We clarified that the full-length VPS55 gene has four transmembrane helixes but the wrong annotation missing the microexon leads to a protein that has only three transmembrane helixes.

Minor point:

1. The labels of species on Figure 1C and Figure 5 are difficult to read when print out.

Response: Thanks for pointing this out. We submitted high-resolution images with the revised manuscript. Please see the independent image files.

2. Figure 6a, the meaning of different colors is unclear. Please also label where is the ATG on those schematics.

Response: We are sorry for confusion here. The different colors indicate different coding exons. To avoid confusion, we changed the color schematics and added more words in legend to explain it in the revised manuscript.

Response to Reviewer #3's Comments to the Author

1. What are the noteworthy results?

The authors identified and analyzed numerous microexons ranging from 1 to 15bp that were previously unannotated, but may be essential to vital functionality in many plant species. They claim that these portions of genomes are not annotated or rather “pervasively misannotated,” as well as describe genetic signals indicating their existence, develop a computational tool for analyzing RNA-Seq data to determine their existence, and apply the tool to a set of genomic data from several plant species.

Response: Thanks to Reviewer #3 for the positive comments.

2. Will the work be of significance to the field and related fields? How does it compare to the established literature? If the work is not original, please provide relevant references.

The work in this paper focuses on microexons that are of the smallest magnitude in terms of length in number of nucleotides. Furthermore, the focus of the manuscript is on plant genomes that have been less studied in terms of the discovery of the microexons. The work in the paper is significant to other fields that seek to identify and examine these microexons in different species, especially at smaller sizes, and can also inform research into specific plant species and their proteins.

Response: Thanks to Reviewer #3 for the positive comments.

3. Does the work support the conclusions and claims, or is additional evidence needed?

The author’s claim of pervasive misannotation among extant plant genetic annotations is slightly too broad; after all, the existing microexons aren’t necessarily misannotated as something else, they are simply not annotated as they have not been discovered. I think misannotation could effectively be replaced with a different word, and I also believe that there should be more focus on providing evidence for this claim if this is the most critical part of the paper in the author’s eyes. Their more profound findings in my opinion are the simple abundance of microexons and the technique they provide to discover these. Unfortunately the methods section of the draft does not really give a clear outline of the overall novel algorithm that is used for this identification; instead the authors opt to describe textually the many different existing software programs that they chained together. In its current form I would suggest that the paper be revised to include a clearer outline of the method that is being proposed by the authors with a potential flow diagram.

Response: We are sorry for the confusion. Here, we adopted the term “misannoation” to refer to the overall accuracy of microexon annotation in the whole gene’s annotation; that is, a microexon is either correctly annotated or incorrectly annotated (i.e., ‘misannotated’) for some reason. In effect, this term is meant to cover all types of errors related to the ME: false positives (an annotated ME is not supported), false negatives (a true ME was not annotated), and situations where an exon boundary is incorrectly identified, such that an ME might be incorrectly annotated as part of a larger exon. We have also found some microexons existing in minor transcripts that were not annotated because these microexons were not discovered. However, according to our splicing analysis of microexons as shown in Supplementary Fig. 3, in most case, the microexons were included in major transcript isoforms (PSI > 0.5 or even close to 1). For genes with major transcript isoforms with microexon inclusion, their annotations of transcripts without microexon-inclusion were incorrect because there were no reads supporting those annotated transcripts (PSI is 0).

For the method, as Reviewer #3's suggestion, we revised the method part. In the manuscript, we have an RNA-seq based microexon discovery flowchart in Supplementary Fig. 1 and a genome-wide microexon prediction flowchart in Fig. 4a. We modified the flowchart diagrams in Fig. 4a with more details. In addition, more detailed description and flowcharts were added to the package in GitHub (<https://github.com/yuhuihui2011/MEPsuite>) as well.

Figure 1b shows a comparison of aligners and their combined uses in terms of the number of reads aligned to at least three exons with a middle exon as a microexon. It is possible that a lot of such reads, especially those aligned to novel microexons, may be incorrectly aligned, which can lead to misinterpretation. I suggest that the authors randomly choose, perhaps 100 such cases (10 cases per species) and perform RT-PCR to calculate false positive rates.

Response: We agree with Reviewer #3's concern. In our microexon discovery pipeline, to reduce the false positives, we used strict criteria for the read mapping, including that the microexon-spanning reads required perfect match to the three parts with no deletions and insertions to avoid incorrect alignment. For validation, we actually already performed RT-PCR experiments to validate 40 cases covering 10 microexons (size ≤ 10 nt) selected from each of four species, i.e. Arabidopsis, soybean, rice and maize. In the revised manuscript, we further conducted simulations to estimate the true positive rate, and we also reported comprehensive comparisons with existing tools with both real RNA-seq data and simulation data. The simulation and real RNA-seq data showed that our method has better true positive rate (Lines 144-155).

4. Are there any flaws in the data analysis, interpretation and conclusions? Do these prohibit publication or require revision?

Again, the methodology seems to be a pipeline of existing bioinformatics tools and statistical tools from various R software packages (i.e., Ballgown, ORFik, etc...). It would be very helpful to the readers if there was a flow diagram clearly depicting the flow of the data from RNA-Seq through the various portions of the proposed pipeline with labeled intermediaries. As it stands, the methodology likely meets the standards in the field, which often include procedures that can be expressed as a conglomeration of various extant packages. However, it is difficult to read the methodology section in this manner, and the methodology is one of the most critical contributions of this manuscript in my opinion, even though the authors suggest that their resulting findings that microexons are pervasively missannotated is more critical to the manuscript.

Response: In the manuscript, we have an RNA-seq based microexon discovery flowchart in Supplementary Fig. 1 and a genome-wide microexon prediction/modeling flowchart in Fig. 4a. In the revised manuscript, we modified the flowchart diagram in Fig. 4a with more details. In addition, more descriptions, more examples, and a detailed flowchart of the whole project were added into the GitHub (<https://github.com/yuhuihui2011/MEPsuite> and <https://github.com/yuhuihui2011/MEPmodeler>).

5. Is the methodology sound? Does the work meet the expected standards in your field?

The authors developed two methods, one being a pipeline for identifying microexons using RNA-seq data and the other, MEP, being a program for predicting microexons using plant genomes. Only the latter program's code is provided on GitHub. Though the authors provide enough detail to reproduce the work that was generated by the former pipeline, I think providing

some scripts with some example sequencing data would be useful so that other researchers could quickly and efficiently annotate other plant genomes processed through the pipeline.

Response: Thanks for Reviewer #3's suggestion. We significantly improved the README files and added some examples to explain how to use the pipeline for annotation in GitHub (<https://github.com/yuhuihui2011/MEPsuite>).

Overall comments: The paper is interesting and relevant, and there is a good set of results and graphics displaying those results. The methodology section leaves something to be desired, however. It would be very nice for a clear separation of the new methodologies introduced by the authors (the prediction tool) and the pipeline constructed with other previously created technologies. To aid with this, I suggest the inclusion of a diagram showing the flow of the RNA-Seq data with intermediary data labeled. In its current form, the paper seems to focus more on the results than on the methodology applied to produce these. I believe that the focus here should be more on the methodology and that the focus on misannotation of microexons in plants should be more appendant to the main findings here, which are critically the methods of discovering very small exons in RNA-Seq data.

Response: We thank Reviewer #3 for the suggestion. In the manuscript, we have a flowchart for the RNA-seq based microexon discovery in Supplementary Fig. 1 and a genome-wide microexon modeling flowchart in Fig. 4a. In the revised manuscript, we modified the flowchart diagrams in Fig. 4a by adding more details. A detailed flow diagram of the whole pipeline was deposited into the GitHub (<https://github.com/yuhuihui2011/MEPsuite>). We also improved the README file in GitHub to provide more information for users.

One minor point is that there are two Supplementary Table 1s, which is confusing. The first instance of Supplementary Table 1 is almost identical to Table 1, so it can be removed.

Response: Thanks for pointing it out. We removed the first Supplementary Table 1 in the revised manuscript.

Response to Reviewer #4's Comments to the Author

The manuscript by Yu et al. describes a pipeline to accurately detect micro-exons (<51 nt) to improve gene annotation. They have further explored the various characteristics of micro-exons and have performed statistical tests to show how those features differ from other exonic features. The implementation of the pipeline, as described in the manuscript, is well organized, clearly presented with useful explanations, figures, and provides useful data for the community. It is particularly valuable for researchers who wish to focus on microexons and its impact on various genome regulatory processes and plant evolution. The approach is comprehensive and the article is a very interesting read. The validations carried out by the authors make their claim even more convincing.

It would be great if the authors could consider the following questions and comments:

The authors have mentioned that they use relaxed thresholds for aligning reads using STAR and OLEgo (line 185). In the methods section, they have included the parameters. For the first pass of STAR, they use `--outSJfilterReads Unique --outSJfilterCountUniqueMin 10 3 3 3 --`

outSJfilterCountTotalMin 10 3 3 3) as parameters. The default parameter setting is (--outSJfilterReads All --outSJfilterCountUniqueMin 3 1 1 1 --outSJfilterCountTotalMin 3 1 1 1). In the second round of STAR, the parameters are set to (--alignSJoverhangMin 20) and the default for this parameter is 5. In both the rounds, it seems like the mapping parameters were more stringent than the default, but the authors claim that they used relaxed parameter settings. Could you please help us understand this difference and what made the thresholds more relaxed?

Response: We are sorry for the confusion. In the revised manuscript, we removed the word of “relax”, and add more words to explain the selection criterion (Lines 140-141). In the original manuscript, we mentioned “relaxed mapping criteria”, but actually, this is not a mapping process. It is a microexon screen or selection process. In order to avoid misunderstanding or confusion, we changed the phrase “relaxed mapping criteria” into “the selection criterion that that allows junction reads to span one or both splice junctions but requiring at least 5 reads spanning each junction”. However, in mapping process, Reviewer #4 is right - we used more stringent criteria than the default to avoid alignment errors. The parameters have been optimized for mapping in plants based on many different tests.

A recently published software, called FINDER, implements STAR and OLego to perform alignment precisely to detect micro-exons and perform gene annotation (<https://bmcbioinformatics.biomedcentral.com/articles/10.1186/s12859-021-04120-9>). Even though the FINDER paper does not explore the characteristics of micro-exons in great depth, it seems to be the one of the first softwares to use OLego to detect micro-exons and therefore warrants a brief discussion in that section of the manuscript.

Response: We thank Reviewer #4's for this suggestion. We included the FINDER software in the Introduction and Discussion in the revised manuscript (Lines 106-107 and 503-504). As we discuss in the manuscript, FINDER merely uses OLego to map unmapped reads after STAR mapping, which is different from our algorithm. Our algorithm interweaves OLego and STAR to utilize each of their advantages to improve microexon detection.

(Comment only) Several plant species have poor gene annotation. I am curious to see how the pipeline will behave if STAR was executed without any support of existing annotations.

Response: STAR performed very poorly without annotations, even worse than HISAT2. Please refer to Fig. 1b and Fig. 4e for this comparison of microexon identification among tools.

What were the reasons that only microexons with canonical splice sites were considered. Would there be a possible method to detect transcripts with internal micro-exons that do not have canonical splice sites?

Response: Yes, this is an important point. During our research, we tested non-canonical splice sites, but we could not find any high-confidence microexons that have non-canonical splice sites. Therefore, we designed our code to focus only on canonical sites, and hence, we did not report any microexons with non-canonical splice sites. In general, non-canonical splice sites are rare (less than 1%) and microexons are also rare (100-300 in total in a genome). Therefore, we do not expect microexons with non-canonical splice sites. Other microexon discovery tools have a similar setup. For example, MicroExonator is even more strict because it only considers U2 introns.

The authors have used Stringtie to assemble the alignments into transcripts. Then they have

used read coverages of introns, exons, and transcripts. Stringtie can be executed to generate coverages for transcripts, but does it also generate read coverages for introns and exons? In addition to that, it will be nice to provide the parameters used to execute Stringtie.

Response: Thanks to Reviewer #4 for this good question. Yes, StringTie can generate read coverages for introns and exons. To implement this function, we added the parameters for using StringTie in the Methods section in the revised manuscript (Lines 612-615). All codes for microexon discovery were deposited into GitHub (<https://github.com/yuhuihui2011/MEPsuite>). Please find all parameters that we used in the file of “golden_map.py.”

It is great that the authors conducted validation tests with a subset of transcripts. RT-PCR tests can pull out the matured mRNA which is devoid of introns. Could you explain how these transcripts were mapped to the reference? From figure 4c, I understand that the red-colored alignment is different from the blue-colored alignment. How did you confirm that the blue-colored alignment (with the 1 nt micro-exon) is the correct alignment (and not the red-colored one), from RT-PCR, Sanger Sequencing, and subsequent mapping? A similar example can be seen in figure 6a, where the red-colored portion in the *P.patens* transcript is a micro exon. In the first example, it is a part of the first exon, but in the second example it is an exon all by itself. I wonder how you would validate it using sequencing. Clarification is appreciated.

Response: We are sorry that we did not clearly describe the results of RT-PCR validation and STAR mapping. Results in Fig. 4c and Fig. 6a were from the STAR mapping of RNA-seq reads, instead of RT-PCR. In Fig. 4c, yes, the blue-colored alignment is the subsequent mapping after we modeled the 1-nt microexon. The original mapping without the 1-nt microexon annotation got only 7 junction reads mapped around the downstream intron of this microexon, but the subsequent mapping with the model of 1-nt microexon got 81 and 88 junction reads supporting the upstream and downstream introns, respectively. The significantly increased number of mapped reads obtained by the subsequent mapping with the model of 1-nt microexon indicates that the model of 1-nt microexon is correct. In Fig. 6a, *P. patens* has two transcripts, one of which has the microexon merged to the first exon (without an upstream intron) and the other has the microexon separated by two flanking introns. The boxes just show the gene structures, but they do not mean that the sequences are identical. In fact, the first exon of the first transcript is in the first intron of the second transcript. For RT-PCR validation, we compared the sequences obtained from Sanger sequencing with the transcript sequences of microexon-tags (microexon and the flanking exons). The sequences from the RT-PCR sequencing must be identical with the predictions to successfully confirm validation.

Is there a software tool available that implements the algorithm discussed that can readily detect micro-exons when provided with the reference and RNA-Seq data?

Response: Yes, the codes for microexon discovery and prediction have been released into GitHub (<https://github.com/yuhuihui2011/MEPsuite> and <https://github.com/yuhuihui2011/MEPmodeler>). For microexon annotation from RNA-seq data, users can use the *golden_map* function in MEPsuite to microexon annotation information in GTF files. For microexon annotation without RNA-seq data, users can use MEPmodeler to predict microexon-tags in genome sequence. For more details, please see GitHub.

In Figure 1a, could the author define the ‘M’ and ‘N’ symbols used in the figure for readers unfamiliar with CIGAR notation.

Response: Thanks to reviewer #4 for pointing it out. We added more words to explain this in the revised manuscript (Fig. 1 legend).

For the explanation of figure 2b, a small note on ISE-hex3-D motif would help to understand the third x-axis label better.

Response: Thanks to reviewer #4 for pointing it out. We added a sentence to describe the ISE-hex3-D motif in the figure legend in revised manuscript (Fig. 2 legend).

The caption for Figure 3b is a bit confusing and hard to read. Rewording is recommended.

Response: We are sorry for the confusion. We revised the caption accordingly.

The word RNA-Seq is inconsistently used in the manuscript ('RNAseq', 'RNA-Seq', and 'RNA-seq').

Response: Thanks to reviewer #4 for pointing this out. We carefully checked the whole manuscript and change all terms to "RNA-seq".

Line 718, the sentence ends by referencing a superscript.

Response: Thanks for pointing it out. We revised it in the manuscript.

Reviewers' Comments:

Reviewer #1:

Remarks to the Author:

The authors clarified most of the points I have raised and the MS has improved.

However, I would still request that the authors clearly discuss the functional relevance of the microexon conservation. That is: that they are not likely to be functionally relevant as ALTERNATIVE exons.

Also, this sentence (and associated claims about microexon loss) should be further clarified: "When microexons get lost, it is most commonly caused by loss of flanking introns from the genes". Microexons per se do not get "lost" under that scenario. It is their definition as microexons. "When microexons get lost" could read: "when the microexon sequences cannot be recognized as microexons"

Reviewer #2:

Remarks to the Author:

In this revised version of manuscript. The author had addressed majority of my concerns. There are still three outstanding questions as I summarized below.

1. Figure 2F is misleading and biased, just by looking at the figure, it suggests the upstream flanking intron of a microexon is removed post-transcriptionally. This is not true as the majority of PTS intron surrounding microexon is spliced out co-transcriptionally (Figure 2d). There is no legend describing Figure 2F properly. Please consider either to remove Figure 2F or to add a parallel route indicating co-transcriptional removal of both flanking introns which is the majority of the cases. To this reviewer, the efficient recognition and removal of flanking intron during transcription even when the exon is so short is actually a surprise. So please present the data and model in an unbiased way.

2. Figure 2d is informative. However, controls are missing here, please show the unspliced ratio calculated from other exons (excluding microexon) as control here. This will provide an unbiased picture how slow of the intron removal when comparing introns flanking microexons to those flanking the ordinary exons.

3. Line 211 to line 216 and in association with Figure 2e. This part is still little confusing. 69 microexons are investigated by looking at reads of polyA CB-RNA fraction. "16 out of 69 microexons had reads supporting the case of sole-downstream-intron splicing and the case of both-flanking-introns splicing but no reads supporting the case of sole-upstream-intron splicing (Fig. 2e), while another four microexons had reads supporting sole-upstream-intron splicing and both-flanking-intron splicing but no reads supporting sole-downstream-intron splicing"

What's the situation of the rest 49 microexons? Please briefly introduce.

The numbers in the figure 2e seems not matching the text above.

4. Refers to my previous comments as copied below. I think the reviewer's response is quite informative, please mention this either in the main text or discussion therefor people understand why the authors only looked the polyadenylated RNA fraction.

B. It is unclear to this reviewer why the author only looked the polyadenylated fraction of chromatin-bound RNA when defining PTS. In this way, the information of co-transcriptional splicing is missed. The author should look at the long reads sequencing data of CB-RNA in which the information of both polyadenylated and nascent RNA are kept. In this way, the author could accurately estimate to what extent these microexons are spliced before and after polyadenylation at the chromatin level.

Response: Thanks to reviewer #2 for this insightful suggestion. We tried using long reads from CB-RNA as suggested, but unfortunately long reads have high sequence errors and could not be used to accurately identify microexons. We have tried different long read correction methods and mapping methods, but could not solve the problem. Importantly, the information of splicing before and after polyadenylation at the chromatin level does not change the main conclusion of this analysis that microexon-flanking introns exhibit an increase of delayed splicing that likely requires PTS.

Reviewer #3:

Remarks to the Author:

I went over the responses and related parts of the revised manuscript. I think that the authors gave satisfactory responses to my questions as well as other reviewers' questions. Overall, I would consider the manuscript to now be in good standing for publication in the Bioinformatics journal.

Reviewer #4:

Remarks to the Author:

I would like to thank the authors for addressing the comments and suggestions in my review. The authors have adequately responded to the review and provided the needed clarifications. The manuscript is substantially improved and will provide clarity to this noteworthy work.

Response to Reviewer #1's Comments to the Author

The authors clarified most of the points I have raised and the MS has improved.

Response: We appreciate Reviewer #1's support.

However, I would still request that the authors clearly discuss the functional relevance of the microexon conservation. That is: that they are not likely to be functionally relevant as ALTERNATIVE exons.

Response: We thank the reviewer for pointing this out. In the revised manuscript, we stated that microexons are not likely to be functionally relevant as alternative exons in the revised manuscript (Lines 494-496).

Also, this sentence (and associated claims about microexon loss) should be further clarified: "When microexons get lost, it is most commonly caused by loss of flanking introns from the genes". Microexons per se do not get "lost" under that scenario. It is their definition as microexons. "When microexons get lost" could read: "when the microexon sequences cannot be recognized as microexons"

Response: Thanks for pointing out this problem. In the revised manuscript, we clarified the statement about the microexon loss in the revised manuscript (Lines 409-416).

Response to Reviewer #2's Comments to the Author

In this revised version of manuscript. The author had addressed majority of my concerns. There are still three outstanding questions as I summarized below.

1. Figure 2F is misleading and biased, just by looking at the figure, it suggests the upstream flanking intron of a microexon is removed post-transcriptionally. This is not true as the majority of PTS intron surrounding microexon is spliced out co-transcriptionally (Figure 2d). There is no legend describing Figure 2F properly. Please consider either to remove Figure 2F or to add a parallel route indicating co-transcriptional removal of both flanking introns which is the majority of the cases. To this reviewer, the efficient recognition and removal of flanking intron during transcription even when the exon is so short is actually a surprise. So please present the data and model in an unbiased way.

Response: We really appreciate these insightful comments and advice. Accordingly, we modified Figure 2F and redrew the model to show most flanking introns are removed co-transcriptionally and some upstream introns are more likely to be detained that require post-transcriptional splicing. We also added more words in the legend to describe this model and modified the description in the main text accordingly in the revised manuscript (Lines 221-224).

2. Figure 2d is informative. However, controls are missing here, please show the unspliced ratio calculated from other exons (excluding microexon) as control here. This will provide an unbiased picture how slow of the intron removal when comparing introns flanking microexons to those flanking the ordinary exons.

Response: Thanks for the constructive suggestion. We added the controls into Fig. 2d and revised the figure legend accordingly to show that the upstream flanking introns of microexons are more likely to be detained.

3. Line 211 to line 216 and in association with Figure 2e. This part is still little confusing. 69 microexons are investigated by looking at reads of polyA CB-RNA fraction. "16 out of 69 microexons had reads supporting the case of sole-downstream-intron splicing and the case of both-flanking-introns splicing but no reads supporting the case of sole-upstream-intron splicing (Fig. 2e), while another four microexons had reads supporting sole-upstream-intron splicing and both-flanking-intron splicing but no reads supporting sole-downstream-intron splicing" What's the situation of the rest 49 microexons? Please briefly introduce. The numbers in the figure 2e seems not matching the text above.

Response: Thanks for pointing out this issue. We added some words to describe the splicing of the rest 49 microexons in the revised manuscript (Lines 215-218). Fig. 2e is just an example of the first case (one of 16 microexons). The numbers in Fig. 2e indicate different types of microexon-spanning gapped reads, and do not indicate different types of microexons. To avoid confusion, we revised the figure legend.

4. Refers to my previous comments as copied below. I think the reviewer's response is quite informative, please mention this either in the main text or discussion therefor people understand why the authors only looked the polyadenylated RNA fraction.

B. It is unclear to this reviewer why the author only looked the polyadenylated fraction of chromatin-bound RNA when defining PTS. In this way, the information of co-transcriptional splicing is missed. The author should look at the long reads sequencing data of CB-RNA in which the information of both polyadenylated and nascent RNA are kept. In this way, the author could accurately estimate to what extent these microexons are spliced before and after polyadenylation at the chromatin level.

Response: Thanks to reviewer #2 for this insightful suggestion. We tried using long reads from CB-RNA as suggested, but unfortunately long reads have high sequence errors and could not be used to accurately identify microexons. We have tried different long read correction methods and mapping methods, but could not solve the problem. Importantly, the information of splicing before and after polyadenylation at the chromatin level does not change the main conclusion of this analysis that microexon-flanking introns exhibit an increase of delayed splicing that likely requires PTS.

Response: Thanks for the insightful suggestion. Accordingly, we added more words in the main text to explain why only the polyadenylated RNA fraction was considered for our analysis in the revised manuscript (Lines 199-200).

Response to Reviewer #3's Comments to the Author

I went over the responses and related parts of the revised manuscript. I think that the authors gave satisfactory responses to my questions as well as other reviewers' questions. Overall, I would consider the manuscript to now be in good standing for publication in the Bioinformatics journal.

Response: We appreciate Reviewer #3's support.

Response to Reviewer #4's Comments to the Author

I would like to thank the authors for addressing the comments and suggestions in my review. The authors have adequately responded to the review and provided the needed clarifications. The manuscript is substantially improved and will provide clarity to this noteworthy work.

Response: We appreciate the nice comments.

Reviewers' Comments:

Reviewer #1:

Remarks to the Author:

Changes done correctly.

Reviewer #2:

Remarks to the Author:

My concerns were addressed by the authors. I am glad to see the work at the current format.